# ENTRYPRUNE: NEURAL NETWORK FEATURE SELECTION USING FIRST IMPRESSIONS

## ABSTRACT

There is an ongoing effort to develop feature selection algorithms to improve interpretability, reduce computational resources, and minimize overfitting in predictive models. Neural networks stand out as architectures on which to build feature selection methods, and recently, neuron pruning and regrowth have emerged from the sparse neural network literature as promising new tools. We introduce EntryPrune, a novel supervised feature selection algorithm using a dense neural network with a dynamic sparse input layer. It employs entry-based pruning, a novel approach that compares neurons based on their relative change induced when they have entered the network. Extensive experiments on 13 different datasets show that our approach generally outperforms the current state-of-the-art methods, and in particular improves the average accuracy on low-dimensional datasets. Furthermore, we show that EntryPruning surpasses traditional techniques such as magnitude pruning within the EntryPrune framework and that EntryPrune achieves lower runtime than competing approaches. Our code is available in the supplementary material.

## 1 INTRODUCTION

Feature selection is a key problem in predictive modelling (Imrie et al., 2022). Since especially in high-dimensional datasets, many features are irrelevant or redundant for predicting the target, it can serve to improve interpretability by highlighting important features, reduce computational resources, or improve predictive performance by reducing overfitting (Li et al., 2018). Its real-world impact is evident—for instance, selecting only 4–14% of features can improve heart attack prediction (Akter et al., 2025), while using just 16–48% of features can enhance cyberattack detection (Umar et al., 2025). Ongoing demand has motivated substantial recent efforts to enhance existing feature selection methods and develop new ones (Theng & Bhoyar, 2024; Pavasovic et al., 2025).

Feature selection algorithms can be categorized into embedded, wrapper, and filter approaches. Embedded methods select features during model training, such as linear regression (Tibshirani, 1996) or neural networks (Lemhadri et al., 2021). Wrapper approaches also work around a specific predictive model, but treat it as a black box with the feature set as a hyperparameter, e.g., via particle swarm optimization (Rostami et al., 2021). Filter approaches select feature sets without being tailored around a predictive model, but using information-theoretic measures. They include, for example, statistical tests of the relationship between the feature and the outcome (Bommert et al., 2020).

Neural networks have a great ability to capture nonlinear relationships and offer many entry points for slightly modifying their architecture or training algorithm to build successful embedded feature selection methods. To decide on the utility of an input neuron, approaches added gates in the input layer (Yamada et al., 2020), added residual connections to the output (Lemhadri et al., 2021), or added gradients with respect to data changes to the loss (Cherepanova et al., 2023).

Feature selection in neural networks translates to aiming for a sparse input layer and is therefore a special case of sparse neural networks (Hoefler et al., 2021). Recently, it was shown that sparse neural network training (Mocanu et al., 2018; Evci et al., 2020) can be adapted to achieve a dominant feature selection performance (Liu et al., 2024; Atashgahi et al., 2024; Sokar et al., 2024). However, in the dynamic sparse regime, competing neurons are active for varying durations, making it challenging to compare them based on real-time metrics. Entry-based pruning addresses this issue by leveling the playing field, allowing regrown neurons to compete more effectively with established neurons. This is achieved by evaluating each neuron based on the change induced upon entering the network.

**All Features**
Downstream Accuracy: 97.92%

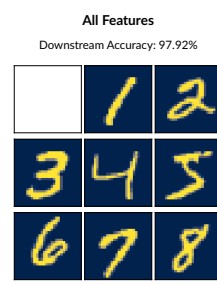

**Fisher (Gu et al., 2011)**
Downstream Accuracy: 74.40%

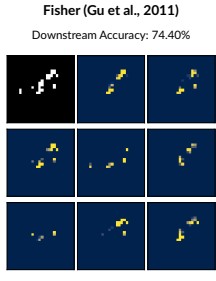

**NeuroFS (Atashgahi et al., 2023)**
Downstream Accuracy: 87.86%

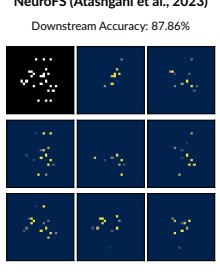

**EntryPrune (ours)**
Downstream Accuracy: 93.04%

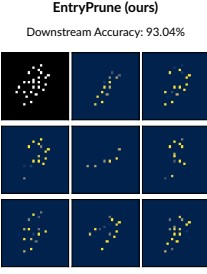

Figure 1: Visualization of feature selection results on MNIST using 25 out of 784 features. Each panel shows the binary mask (top left) of the selected pixels, along with sample digits where only the selected pixels are visible. The downstream accuracy scores (from an SVM classifier) were computed by training on only these selected features and are averaged across five runs (see Appendix E for details). Our approach enhances interpretability by identifying a minimal set of critical features that maintain classification performance. Appendix A includes a similar visualization for CIFAR-10.

In this paper, we introduce EntryPrune, a novel neural network feature selection algorithm implementing dynamic sparsity via random regrowth and entry-based pruning in the input layer of a dense neural network. Our main contributions are:

1. Entry-based pruning, a novel pruning approach that compares neurons based on remembering the initial importance of features when they enter the network. We show that it outperforms established pruning methods such as magnitude pruning in the context of our algorithm.

2. The EntryPrune feature selection algorithm, which has two key hyperparameters that allow it to adapt to the characteristics of the dataset used. A variant can adjust the input layer size at runtime, reducing sensitivity to one of its hyperparameters.

3. An extensive experimental evaluation of the approach, demonstrating that it outperforms state-of-the-art methods such as NeuroFS and LassoNet on 9 out of 13 diverse datasets. An illustration is given in Figure 1.

The paper is structured as follows: We start with a review of related work, focusing on neural network-based methods. Then, we present the EntryPrune algorithm and its extension with an adaptive input layer. Next, we report extensive experiments evaluating our approach. Finally, we include auxiliary analyses on design parameters and computational efficiency.

## 2 BACKGROUND AND RELATED WORK

We introduce the feature selection problem in the context of neural networks and review prior solutions. Most methods slightly modify a dense network architecture or its loss function. More recently, successful approaches have drawn from sparse neural network frameworks.

**Feature selection in neural networks.** We consider the task of selecting a set of $K$ features that are most valuable for making accurate predictions in a supervised learning setting. The feature selection problem can be formulated as: $\arg\min_{S \subset \mathcal{F}, |S| = K} \mathcal{L}(S)$, where $\mathcal{F}$ is the full feature set, $|S| = K$ constrains the selection to exactly $K$ features, and $\mathcal{L}(S)$ is the loss function of a downstream learner using only the selected features $S$. This is an NP-hard problem and becomes intractable in high-dimensional settings (Yamada et al., 2020; Theng & Bhoyar, 2024), which can be illustrated through a practical example: when selecting a set of 25 features from the MNIST dataset (Figure 1), there are approximately $10^{47}$ possible combinations – making exhaustive search computationally infeasible. For neural networks, the task of feature selection translates to implementing an effective $L^0$ regularization of first layer weights $\mathbf{W}^{(1)}$. We say that $K$ neurons are active when

$$\left| \{ i \mid L^0(\mathbf{W}^{(1)}_{i.}) > 0 \} \right| = K, \tag{1}$$

where $\mathbf{W}^{(1)}_{i.}$ is the vector of outgoing first layer weights from input neuron $i$. The related work discussed below uses various approximations to address this challenge.

**Dense neural networks.** There are several methods embedded in dense neural networks for feature selection. A common property is that the number of active neurons is not strictly enforced before model convergence. Instead, selection is gradual, starting with a full input layer of $N$ neurons and reducing active neurons during training. This approach makes it easier to identify complex interactions between features, at the cost of increased computational complexity. CancelOut (Borisov et al., 2019) applies a simple deterministic scaling to the input features in a dedicated initial layer. A trainable weight vector passes through a sigmoid, and L1 and L2 regularization suppress irrelevant features. Stochastic gates (Yamada et al., 2020) approach the $L^0$ regularization by adding a gate to each input layer neuron. For each gate, a trainable parameter controls the probability of a feature being active. The LassoNet (Lemhadri et al., 2021) adds a residual connection from each input layer neuron to the network output. The absolute sizes of these $N$ residual weights are added to the loss function and for each feature $i$ individually represent a bound on the size of the corresponding first layer weights, $||\mathbf{W}_{i\cdot}^{(1)}||_1$. A less invasive approach is DeepLasso (Cherepanova et al., 2023), which adds the gradient with respect to changes in the input data to the loss function. This encourages the network not to use some features during training, rendering the corresponding input neuron inactive.

**Sparse neural networks.** Sparse neural networks maintain a large fraction of zero-valued weights throughout the network to reduce memory requirements or computational cost (Hoefler et al., 2021). Sparsity can be introduced either by pruning a trained dense model or by maintaining a sparse structure throughout training, as in Dynamic Sparse Training (DST) (Nowak et al., 2023). Various metrics have been proposed to guide structured pruning of neurons, including weight magnitudes (Evci et al., 2020; Mocanu et al., 2018) or mathematical approximations of loss change (Molchanov et al., 2019; Lee et al., 2019). In DST, neurons are typically regrown either randomly (e.g., Mocanu et al., 2018) or based on the magnitude of adjacent gradients (Evci et al., 2020).

GradEnFS (Liu et al., 2024) is one approach utilizing DST for feature selection. Similar to DeepLasso, it measures the importance of neurons based on how sensitive the loss is to changes in the input neurons. After the model converges, it selects the top $K$ features based on neuron importance. NeuroFS (Atashgahi et al., 2023) extends DST approaches (Mocanu et al., 2018; Evci et al., 2020) to the input layer. Input neurons are pruned after each epoch based on the magnitude of their outgoing connections, $||\mathbf{W}_{i\cdot}^{(1)}||_1$. To regrow an input neuron, NeuroFS calculates the absolute gradients of all currently pruned first layer weights. Neurons are then regrown based on the largest absolute gradient among their adjacent weights. During training, the number of active neurons in the input layer is continuously reduced. After training, the input neurons with the largest outgoing connections among the remaining active neurons are selected.

We identify three opportunities for improvement in existing work: (1) Current pruning metrics compare regrown neurons to ones active for longer, giving them unequal time to accumulate metrics. We propose a metric that instead evaluates features based on their initial short-term impact. (2) Gradient-based regrowth favors features with large initial gradients. But features lacking a linear correlation with the target show gradients similar to noise features (see Appendix B). Consequently, we favor random regrowth for its ability to better uncover interactions and its reduced computational cost by omitting gradient calculations. (3) Prior work uses either fully sparse or fully dense networks. We propose a dynamically sparse input layer with a dense body, where only the input layer is sparse. Since modern GPUs are architected and optimized for dense matrix-matrix multiplication, existing sparse kernels surpass dense ones only when the sparsity level is high (>70%) – consequently, with an equivalent parameter count, sparse kernels remain well short of the performance achieved by dense kernels (Gale et al., 2020; Okanovic et al., 2024).

## 3 THE ENTRYPRUNE ALGORITHM

We propose the EntryPrune algorithm for supervised feature selection using neural networks. This section walks through the pseudocode in Algorithm 1 and explains its rationale. The core loop of entry-based pruning and random regrowth is illustrated in Figure 2. EntryPrune is implemented using PyTorch (Paszke et al., 2019) and is available as a Python package in the supplementary material.

**Architecture and initialization.** The algorithm uses a multi-layer perceptron (MLP) with a feed-forward architecture and is integrated into the backpropagation training using the Adam optimizer

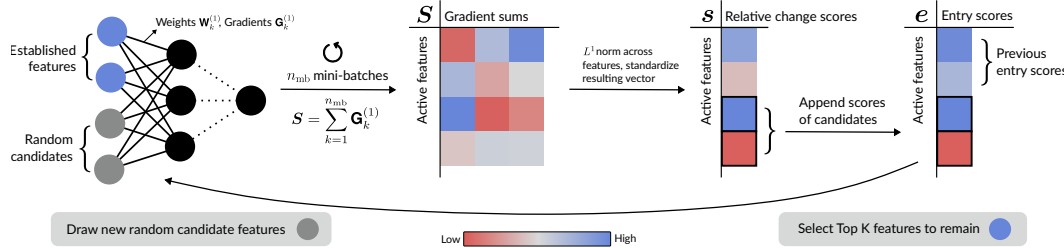

Figure 2: Entry score calculation in EntryPrune, Algorithm 1. The network's input layer includes $K$ selected features plus extra candidates. Over several mini-batches, set by the hyperparameter $n_{\mathrm{mb}}$, first layer gradients $\mathbf{G}_k^{(1)}$ are added in a matrix $\mathbf{S}$. We then compute the $L^1$ norm for each input neuron and standardize the resulting vector to get relative change scores $s$. Candidate scores are entered into entry score vector $e$. Features with the top $K$ entry scores stay; others are randomly regrown. Candidate feature weights are reinitialized before training continues.

---

**Algorithm 1** EntryPrune

1: **Input.** Dataset with $N$ features, number of selected features $K$, number of first hidden layer neurons $n_{\mathrm{hidden}}$. Hyperparameters: Ratio of candidate features $c_{\mathrm{ratio}}$, number of mini-batches $n_{\mathrm{mb}}$
2: **Initialize.** Number of candidate features $K_c = \mathrm{round}(c_{\mathrm{ratio}}(N - K))$. Network with input layer size $K + K_c$. Randomly choose features to populate the input layer, $I_{\mathrm{input}} = I_{\mathrm{cands}} = \mathrm{Rand}(\{1, \ldots, N\}, K + K_c)$. Score vector $s \in \mathbb{R}^{K+K_c}$, entry score vector $e$ is initialized as an empty vector. First layer gradients $\mathbf{G}^{(1)}$ and gradient sum matrix $\mathbf{S}$: $\mathbf{G}^{(1)}, \mathbf{S} \in \mathbb{R}^{(K+K_c) \times n_{\mathrm{hidden}}}$
3: **while** training not stopped **do**
4:     $\mathbf{S} = 0$
5:     **for** $n_{\mathrm{mb}}$ mini-batches **do**
6:       *Feed-forward step and backpropagation using a mini-batch of data*
7:       $\mathbf{S} = \mathbf{S} + \mathbf{G}^{(1)}$
8:     **end for**
9:     $s_i = \sum_{j=1}^{n_{\mathrm{hidden}}} |\mathbf{S}_{ij}|$ for $i \in \{1, \ldots, K + K_c\}$
10:    Normalize to obtain relative change scores $s = (s - \mathrm{Mean}(s))/\mathrm{SD}(s)$
11:    Extend the entry score vector $e$ with the relative change scores of the candidate neurons
12:    Select the top $K$ scoring features from $e$, indexed by $I_{\mathrm{top}}$, and remove all other entries
13:    Draw new candidates $I_{\mathrm{cands}} = \mathrm{Rand}(\{1, \ldots, N\} \setminus I_{\mathrm{top}}, K_c)$
14:    Update features that populate the input layer $I_{\mathrm{input}} = I_{\mathrm{top}} \cup I_{\mathrm{cands}}$
15:    Initialize candidate first layer weights $\mathbf{W}_{\mathrm{cands.}}^{(1)} = U(-10^{-8}, 10^{-8})$. Initialize the optimizer
16: **end while**

---

(Goodfellow et al., 2016; Kingma & Ba, 2015). This necessitates the adoption of the hyperparameters of learning rate, batch size, and number of hidden layers and their sizes. The size of the input layer is based on the desired number of selected features $K$ plus a percentage $c_{\mathrm{ratio}}$ of the remaining features, $K_c$, which will be referred to as candidates. We discuss the tradeoffs involved in choosing the input layer size via the $c_{\mathrm{ratio}}$ hyperparameter in a paragraph below.

**Relative change scores.** Steps 5-10 calculate the relative change scores $s$, where gradient sums for each input neuron are aggregated and normalized to reflect their relative contribution across the last $n_{\mathrm{mb}}$ mini-batches (see also Figure 2). Instead of gradient sums, one could also use weight changes as a relative change metric in Steps 5-8, which we compare in an ablation study in Section 4.2.

**Entry-based pruning.** The entry score $e$ is the relative change score of each feature after the first $n_{\mathrm{mb}}$ mini-batches a feature is in the network. We therefore only add entries for regrown candidates to the entry score vector $e$ in Step 11. The entry scores are then used in Step 12 to select features that remain in the network, and $e$ is updated accordingly. This way, EntryPrune accounts for that features might be in the network for different time spans. We are not aware of other pruning techniques that account for this: For example, magnitude pruning (Atashgahi et al., 2023) or the importance score by

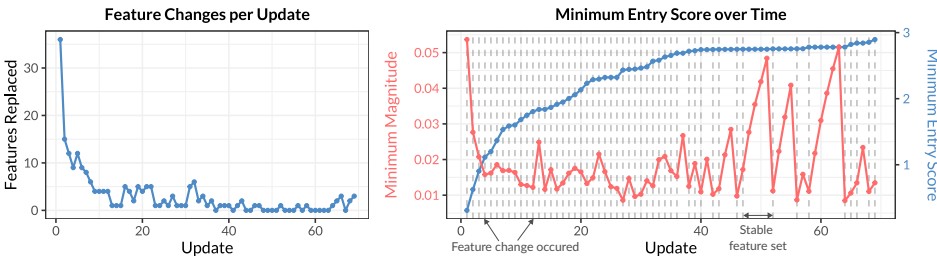

Figure 3: EntryPrune runtime metrics. Left: Number of changes to the top $K$ features over time. Right: Minimum entry score (blue) and minimum absolute first layer weight magnitude (red) among top $K$ features. Changes become less frequent as training progresses and the minimum entry score increases. Minimum weight magnitudes increase during stable phases (e.g., updates 47–51).

Molchanov et al. (2019) would compare metrics of newly added features with those of features that have been in the network longer. We compare the performance of entry-based pruning with these approaches in Section 4.2.

**Random Regrowth.** New candidate features are randomly sampled from the remaining features (Step 13) to form the new input layer together with previous top features (Step 14). To avoid introducing noise from standard initialization methods, which typically use larger weight magnitudes, the weights of candidate features are reinitialized to very small random values, uniformly sampled from $[-10^{-8}, 10^{-8}]$ (Step 15). This also ensures symmetry breaking during training (Goodfellow et al., 2016). Randomly regrowing weights is common in DST (Nowak et al., 2023). For feature selection, it is particularly promising because it allows features to incrementally prove their relevance: instead of relying on gradient signals for inclusion (Atashgahi et al., 2023), candidates are randomly reselected and evaluated across multiple mini-batches. This benefits features contributing to complex, non-linear patterns (see our demonstration in Appendix B).

**Rationale.** EntryPrune alternates between two coupled processes — continuous weight optimization (SGD) and discrete, history-dependent mask updates produced by entry-based pruning and random regrowth. Each rotation resets a subset of first-layer weights to near-zero, changing the parameter space and the effective loss surface at every step. Consequently, the optimization problem becomes *non-stationary* and *bi-level*: the inner level updates weights given a fixed mask, while the outer level updates the mask as a function of the weight trajectory. Standard SGD convergence analyses, which assume a fixed, smooth objective and unbiased gradients, no longer apply. The only rigorous result we are aware of (Alistarh et al., 2018) covers *gradient sparsification* under convex assumptions and cannot be extended to our setting. Hence, the analysis would require new tools for non-convex, time-varying bi-level optimization and is therefore left for future work.

We show an example of the stabilization of the entry score vector during runtime in Figure 3. As high-quality features are established, less important features are increasingly unlikely to enter the network, leading to a more stable feature set. The figure highlights a key advantage of our entry score mechanism over magnitude pruning: as the feature set stabilizes, the magnitude threshold for new candidates grows over time (due to increasing magnitudes of active features), while the entry score remains constant. This allows high-quality features to enter the network even after prolonged training, which would be suppressed under magnitude-based criteria.

**Input layer size.** Balancing the input layer size $c_{\text{ratio}}$ reflects an exploration–exploitation tradeoff. Smaller input layers favor exploitation by stabilizing learning and refining already promising features. Larger input layers promote exploration by sampling more candidate features simultaneously, increasing the chance of uncovering feature interactions. However, they also introduce more noise: the downstream network must continually adjust to frequent weight resets affecting a larger portion of the input layer. As detailed in Appendix C, this tradeoff manifests in both feature selection stability and model performance. To navigate it automatically, we introduce EntryPrune flex in Appendix D, which dynamically adapts $c_{\text{ratio}}$ during training.

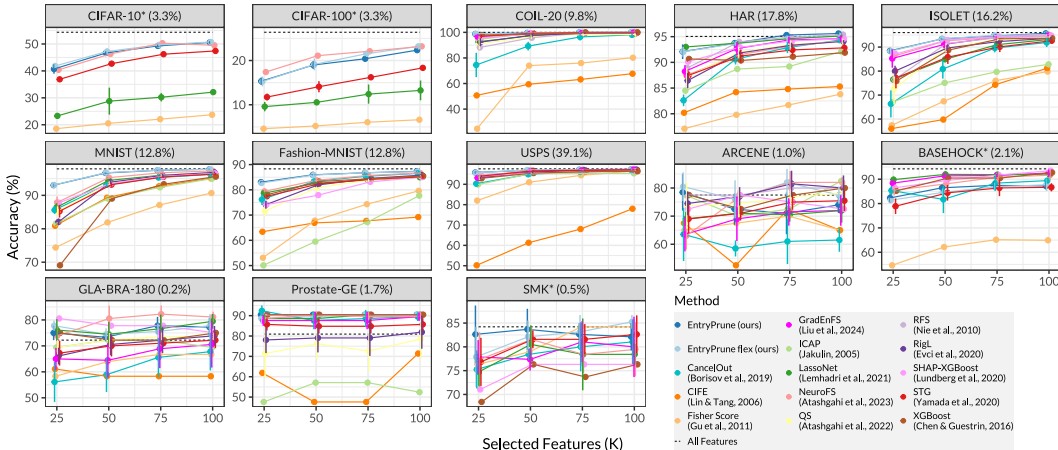

Figure 4: Resulting accuracy for the studied methods by dataset and number of selected features $K$ using the SVM downstream learner. "All Features" is the accuracy using all features in the dataset. Each point shows the mean accuracy across five runs, with error bars indicating the standard deviation. The percentage shown in parentheses after each dataset name indicates the proportion of features that $K = 100$ corresponds to, relative to the full feature set. Datasets marked with an asterisk were evaluated with a limited set of baseline methods (see Appendix E), while baseline results for the other datasets are reproduced from Atashgahi et al. (2023). Results for all downstream learners are shown in Appendix F.

## 4 EXPERIMENTS

We demonstrate the potential of EntryPrune for feature selection through a comprehensive experimental evaluation across 13 diverse datasets and a range of additional analyses. We categorize datasets as *long* if they have more cases than features, and *wide* otherwise. To conserve computational resources, we replicate the experimental setup of Atashgahi et al. (2023), which is feasible for nine datasets.[1] We compare our results with those of nine state-of-the-art baseline methods reported in their work. The experimental protocol includes feature selection using the candidate approaches and subsequent predictive performance measurement using a set of downstream learners. Experimental details, including dataset characteristics, baseline methods, and model configurations, are provided in Appendix E. Code for replicating the main experiment is available in the supplementary material.

### 4.1 RESULTS

Figure 4 presents a comparison of the accuracies achieved using our methods ("EntryPrune" and "EntryPrune flex") against the top baseline methods for the SVM downstream learner. The average accuracy by dataset is shown for all methods in Figure 5. Detailed results for each dataset, method, and value of $K$ are provided in Appendix F.

According to the results, our methods consistently outperform the baseline methods for long datasets (first eight panels in the plots). In particular, they achieve notable improvements on ISOLET, MNIST, and FASHION-MNIST. For MNIST, our flex variant reaches an average accuracy of 96.3%, significantly surpassing the best previously reported result of 94.3%. On CIFAR-100, our approaches perform slightly below NeuroFS, likely due to architectural differences. As demonstrated in Appendix G, EntryPrune surpasses competing methods on this more complex dataset when paired with a larger architecture.

---

[1]This includes code for data preprocessing, train-test split, and downstream learners, which is available at https://github.com/zahraatashgahi/NeuroFS. The performance of the downstream learners using all features was compared with the reported values to ensure accurate replication of the experiment setup. As detailed in our supplementary material, this was unsuccessful for the BASEHOCK and SMK datasets, which are therefore run separately alongside the added CIFAR-10 and CIFAR-100 datasets.

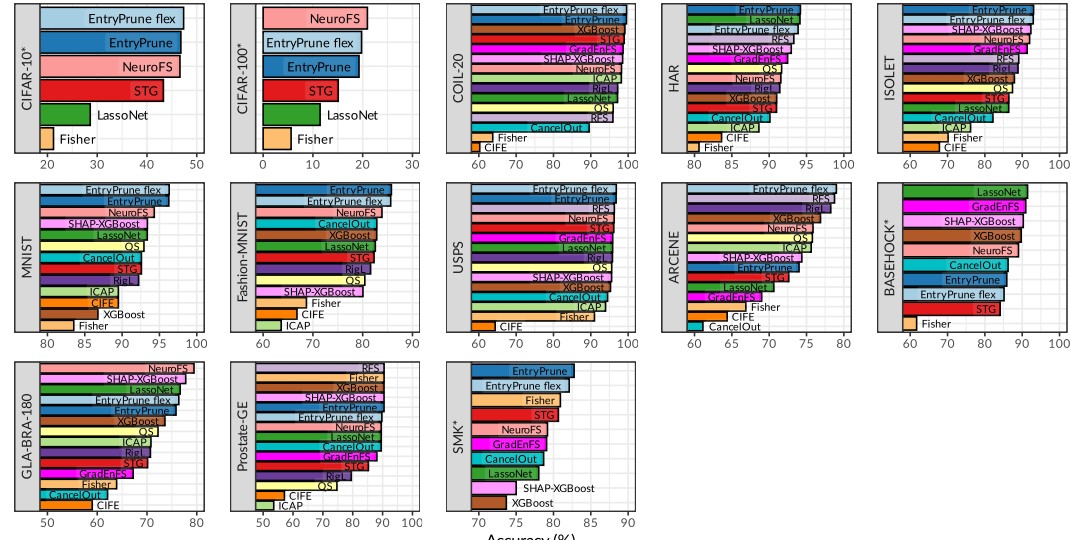

Figure 5: Average accuracy (across all values of $K$) by dataset for the studied methods using the SVM downstream learner. Our proposed methods are "EntryPrune" and "EntryPrune flex". Datasets marked with an asterisk were evaluated with a limited set of baseline methods (see Appendix E), while baseline results for the other datasets are reproduced from Atashgahi et al. (2023). Results for all downstream learners are shown in Appendix F.

For the wide datasets (last five panels in the plots), performance is generally comparable to the baselines. Our approach yields competitive results for the SMK, ARCENE, and PROSTATE-GE datasets, while results for GLA-BRA-180 and BASEHOCK are slightly lower than those of the top-performing baselines. The EntryPrune and EntryPrune flex variants perform similarly across most datasets, with a more pronounced difference observed for the ARCENE dataset, where the EntryPrune approach trails the strongest baselines. However, as visible in Figure 4, the higher standard errors for the wide datasets imply lower separability of the methods' performances.

We also evaluated two additional downstream learners, KNN and ET, for $K = 50$ selected variables (see Tables 5 and 6 in Appendix F). The results are very similar to those obtained with the SVM classifier, indicating that the selected feature sets are valuable across multiple downstream learners.

## 4.2 ADDITIONAL ANALYSES

In this section, we highlight some additional aspects to give a more complete picture of EntryPrune. We include a comparison of the computational efficiency with similar methods, an ablation study of the impact of the chosen change metric, and an investigation of the impact and feasible ranges of hyperparameters. Additionally, we provide analyses of feature selection stability in Appendix C and stopping criteria in Appendix H.

**Computational efficiency.** We examine the comparative computational costs with two other approaches, NeuroFS and LassoNet. Both are well-performing sparse and dense neural network based methods, respectively. One apparent disadvantage of our approach is that, since candidate features are chosen randomly, it generally requires more training epochs than gradient-based approaches to ensure that all linearly correlated features have a chance to enter the network (see also Appendix B). This motivates comparing the overall runtime of the approaches.

We measure the wall-clock time for selecting $K = 50$ features, using two wide and two long datasets, with settings otherwise as in the main experiment. For NeuroFS, we use the setup from the original publication: a 3-layer sparse MLP with 1000 neurons in each layer, limiting the training epochs to 100. For LassoNet, we use the same MLP architecture as for EntryPrune, i.e., one hidden layer with

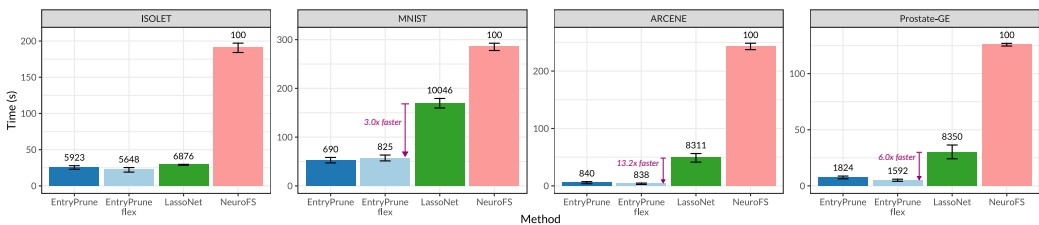

Figure 6: Wall-clock run time for the studied methods by dataset, measured over the entire training duration. All configurations use $K = 50$ selected features and are repeated five times. The error bars indicate the standard deviation across five runs. Mean training epochs for each method and dataset are annotated above the bars.

100 neurons. We keep all other settings at the LassoNet package defaults.[2] Each configuration is run five times.

The results are shown in Figure 6. The EntryPrune and EntryPrune flex approaches have comparable runtimes, both demonstrating significantly greater efficiency than NeuroFS across the studied datasets. Additionally, EntryPrune is more efficient than LassoNet in the studied configurations. In terms of training epochs, EntryPrune achieves similar efficiency while requiring fewer epochs than LassoNet.

We also assessed the overhead introduced by EntryPrune relative to standard dense MLP training. For long datasets, the overhead remains minimal—between 5% and 10%. For wide datasets, the overhead is higher (approximately 220%), primarily due to more frequent feature rotations (20× more often) and a larger pool of candidate features. This trade-off enables EntryPrune to retain flexibility and perform thorough feature exploration, while still maintaining a low absolute runtime across diverse dataset types.

**Ablation study: Pruning metrics.**   We compare the performance of EntryPrune under different change metrics, both with and without an entry score, to classical pruning approaches. Among the change metrics, we evaluate the gradient sums used in EntryPrune against alternatives based on weight changes or magnitude. In both cases, the computation of $S$ is modified just before Step 9 of Algorithm 1. For weight changes, we set $S = \mathbf{W}^{(1)} - \mathbf{W}^{(1)}_{\text{old}}$, where $\mathbf{W}^{(1)}_{\text{old}}$ denotes the first layer weights from the time of the last rotation. For magnitude, we simply set $S$ equal to the first layer weights, $S = \mathbf{W}^{(1)}$. Each of these three configurations is evaluated both with and without using an entry score. For instance, the magnitude approach without an entry score corresponds to standard magnitude pruning, where features are retained based on having the highest absolute sums of first-layer weights at each rotation. In addition, we evaluate the pruning method of Molchanov et al. (2019) (Equation 8 in their paper), which does not utilize an entry score. To implement this method, we accumulate an importance score for each weight after every mini-batch using $(gw)^2$, where $g$ is the gradient and $w$ is the weight. At rotation time, we select the top $K$ features based on the sum of these importance scores corresponding to each feature. We use four datasets, two long and two wide, and $K = 50$ selected features, keeping all other properties the same as in the main experiment.

Figure 7 shows the results. For the long datasets (left two panels), the gradient sums and weight changes perform similarly, surpassing the performance of absolute weights. For the wide datasets (right two panels), the gradient sums show superior performance, while the other two approaches exhibit similar effectiveness. In summary, under the studied configurations, gradient sums are the most effective metric for measuring relative change within the EntryPrune algorithm.

**Impact of hyperparameters.**   We investigate the role of the hyperparameters $c_{\text{ratio}}$ and $n_{\text{mb}}$. Generally, $c_{\text{ratio}}$ determines the percentage of features included in the network in addition to the $K$ selected features, while $n_{\text{mb}}$ specifies the number of mini-batches after which scores are computed and features are rotated. We use both long and wide datasets—HAR, ISOLET (long), and ARCENE (wide)—with $K = 25$, and keep all other properties consistent with the main experiment. We let $c_{\text{ratio}}$

---

[2]The LassoNet package is available at https://github.com/lasso-net/lassonet.

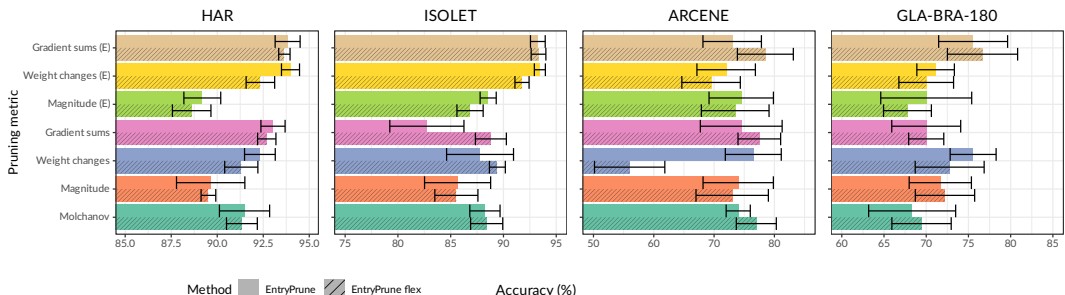

Figure 7: Resulting accuracy for the studied pruning metrics by EntryPrune method and dataset for $K = 50$ selected features using the SVM downstream learner. Metrics annotated with "(E)" are using an entry score. The error bars indicate the standard deviation across five runs.



Figure 8: Accuracy using EntryPrune feature selection by hyperparameters $c_{\text{ratio}}$ and $n_{\text{mb}}$ for three datasets and $K = 25$ selected features. Each point represents the average of three runs. As a visual guide, a Gaussian Process interpolation is shown over the hyperparameter space.

vary between 0.01 and 1 and $n_{\text{mb}}$ between 1 and 150. As studied hyperparameter sets we include the two configurations from our experiment: ($c_{\text{ratio}} = 0.2$, $n_{\text{mb}} = 100$) for the long datasets and ($c_{\text{ratio}} = 0.5$, $n_{\text{mb}} = 5$) for the wide datasets. Additionally, we include the four corners of the hyperparameter space and draw 40 pseudo-random sets of configurations from a Halton sequence. Each resulting configuration is run three times, and the accuracy is averaged.

The results are illustrated in Figure 8, where the impact of the hyperparameters is shown via a Gaussian Process interpolation. As a visual guide, the fit smooths the observed accuracy values across the hyperparameter space, providing a continuous view of the relationship between $c_{\text{ratio}}$ and $n_{\text{mb}}$ for the datasets. For the long datasets HAR and ISOLET (left and center panels), the combination of low $c_{\text{ratio}}$ and high $n_{\text{mb}}$ yields strong results. In contrast, for the ARCENE dataset (right panel), configurations with low $n_{\text{mb}}$ generally perform well. A combination of low $c_{\text{ratio}}$ and higher $n_{\text{mb}}$ may also be effective. In summary, as a rule of thumb, long datasets perform best with low $c_{\text{ratio}}$ and high $n_{\text{mb}}$, while wide datasets perform well with moderate $c_{\text{ratio}}$ and low $n_{\text{mb}}$; the configurations used in our experiments provide practical starting points for tuning.

## 5  CONCLUSION

In this paper, we introduce EntryPrune, a novel feature selection algorithm designed to enhance interpretability, reduce computational demands, and mitigate overfitting in predictive models. Its dynamically sparse input layer employs EntryPruning – a novel approach that evaluates features based on the relative change they induce upon entering the network, enabling fairer comparison among competing neurons. Extensive experiments demonstrate that our method, along with an extension featuring an adaptive input layer, consistently outperforms state-of-the-art techniques on datasets with more cases than features. For datasets with more features than cases, its performance is comparable to previous approaches. While the adaptive version has theoretical advantages and performs better on one dataset, the base algorithm stands out for its simplicity and competitive performance in most scenarios.

## REPRODUCIBILITY STATEMENT

We include the source code of our method in the form of a Python package, as well as code to reproduce the main experiment results, in the supplementary material.

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

## A   FEATURE SELECTION VISUALIZATION FOR CIFAR-10

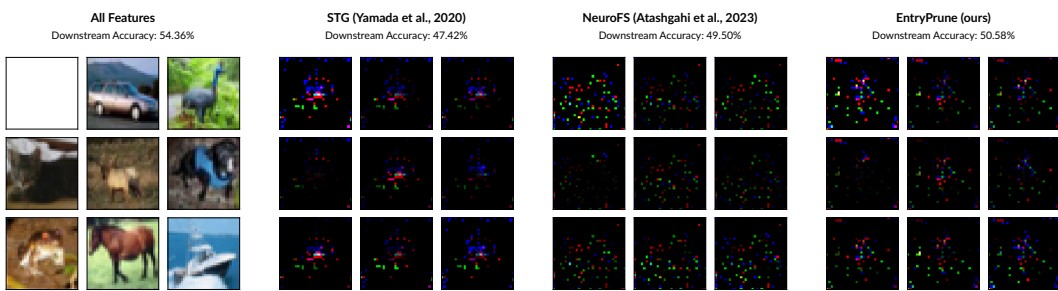

Figure 9: Visualization of feature selection results on CIFAR-10 using 100 out of 3072 features. Each panel shows the binary mask (top left) indicating the selected features (combinations of pixel and color), along with sample images where only the selected features are visible. The downstream accuracy scores (obtained using an SVM classifier) were computed by training on only these selected features and are averaged across five runs (see Appendix E for details).

## B   GRADIENT-BASED REGROWTH AND INTERACTION FEATURES

We investigate a limitation of gradient-based regrowth in the context of interaction features using a toy example. We define linear features as those that relate linearly to the target, and interaction features as those that are uncorrelated with the target individually but contribute to predictions when combined with other interaction features through an XOR (exclusive-or) logic.

Gradient-based regrowth, as used in Evci et al. (2020) and Atashgahi et al. (2023), selects candidates for regrowth based on the highest absolute gradient of adjacent weights. In this toy example, we examine how this metric behaves for interaction features. We generate a dataset with 20 features: 6 linear, 6 interaction, and 8 noise features. EntryPrune is run, and the gradient metric is computed for all candidate features immediately after their weights are reset, since this is the point at which candidates would typically be selected for regrowth in upcoming mini-batches. The resulting average rankings are recorded and presented in Figure 10. In the left panel, we observe that linear features are

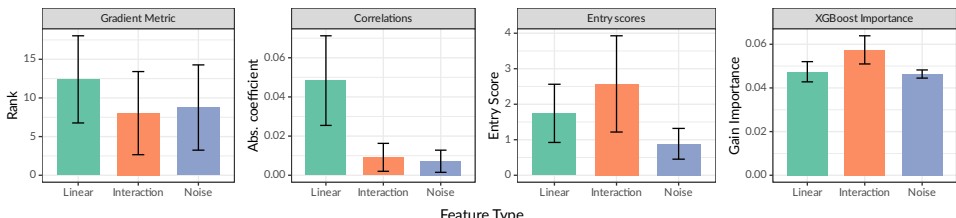

Figure 10: Metrics for a toy dataset using linear, interaction, and noise features. The left panel shows the mean Gradient Metric used in gradient-based regrowth, measured during EntryPrune's runtime. The other panels show the mean correlation of features with the target, the mean Entry Score after training EntryPrune, and the mean gain (XGB importance measure). The error bars indicate the standard deviation across five runs.

ranked highest according to the gradient metric, while interaction features are ranked similarly to noise features. This implies that interaction features would be selected for regrowth at roughly the same rate as noise features. The remaining panels serve to validate the toy dataset: linear features show correlation with the target, while interaction and noise features do not; however, interaction features receive the highest importance scores, both in terms of the entry scores and XGBoost feature importance (Chen & Guestrin, 2016).

One likely explanation for this observation is that the initial gradient immediately after a feature is added to the network does not capture complex interactions. It may primarily reflect simple

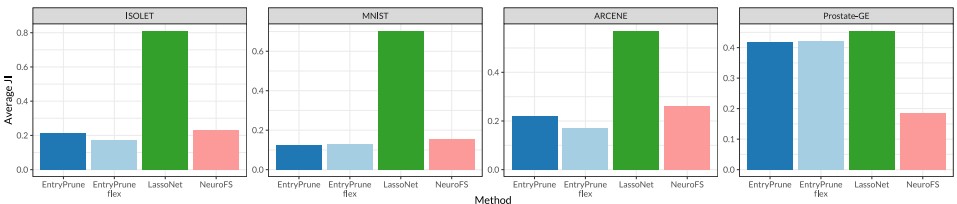

Figure 11: Average Jaccard Index of the selected feature sets by dataset. Higher values represent higher feature set stability between runs. All configurations use $K = 50$ selected features and five runs.

correlations with the target, whereas the network requires more mini-batches to exploit interactions between features. As a result, random regrowth may outperform gradient-based regrowth when it comes to recovering interaction features.

## C    FEATURE SELECTION STABILITY

In this section, we want to explore the impact of random regrowth on feature selection stability, i.e. how similar the selected feature sets are across runs. We first conduct a detailed analysis on the MNIST dataset and then compare feature selection stability with other methods across multiple datasets.

**Input layer size and stability.**    A key factor influencing EntryPrune's function is the $c_{\text{ratio}}$ parameter, which determines the input layer size. A smaller input layer may reduce stability since important interacting features are less likely to appear together during random regrowth. Conversely, a larger input layer (e.g., $c_{\text{ratio}} = 0.8$) increases the likelihood of co-occurrence but introduces more noise, potentially hindering training due to frequent resets of a larger number of weights.

To investigate this, we assess the effect of different $c_{\text{ratio}}$ values on the MNIST dataset using $K = 50$ selected features, three $c_{\text{ratio}}$ settings, and 20 runs each. Feature selection stability is measured by averaging the Jaccard indices (JI; for an overview, see Khaire & Dhanalakshmi, 2022) of selected feature sets. Table 1 presents the results. The findings confirm our theoretical expectations: increasing

Table 1: Jaccard Index of the selected feature sets by $c_{\text{ratio}}$. Higher values represent higher feature set stability between runs. All configurations use $K = 50$ selected features and 20 runs.

| $c_{\text{ratio}}$ | JI | EntryPrune Accuracy $\pm$ SD | SVM Accuracy $\pm$ SD |
|---|---|---|---|
| 0.2 | 0.13 | $95.80 \pm 0.43$ | $96.84 \pm 0.16$ |
| 0.5 | 0.15 | $93.95 \pm 1.05$ | $96.71 \pm 0.14$ |
| 0.8 | 0.21 | $91.50 \pm 1.87$ | $96.06 \pm 0.22$ |

$c_{\text{ratio}}$ leads to greater overlap in selected feature sets but also results in decreased EntryPrune and SVM accuracies with increased variance. Reducing the randomness of regrowth improves feature selection stability but at the cost of lower and less stable model accuracy. This highlights the exploration-exploitation tradeoff: a larger input layer increases noise, preventing the EntryPrune from refining optimal solutions. Consequently, we either obtain slightly worse but consistent feature sets or better sets with reduced overlap across runs.

**Comparison with other methods.**    To understand the implications of this tradeoff, we assess how our chosen $c_{\text{ratio}}$ settings (0.2 for long and 0.5 for wide datasets) influenced feature selection stability in our main experiment. We analyze four datasets—two long and two wide—with $K = 50$ selected features. We compare EntryPrune with two similar neural-network-based approaches, LassoNet and NeuroFS. The results are illustrated in Figure 11. The results indicate that LassoNet generally achieves much higher stability than NeuroFS and EntryPrune, which exhibit similar performance. This discrepancy may stem from initialization variability: LassoNet consistently transitions from a dense

starting point to a regularized endpoint, whereas NeuroFS and EntryPrune begin with randomized sparsity patterns. Furthermore, EntryPrune demonstrates slightly lower JI values than NeuroFS, likely due to differences in feature selection mechanisms—gradient-based selection (NeuroFS) versus random regrowth (EntryPrune). Input neurons with high gradients may be more consistently selected across runs in NeuroFS. In summary, our analysis shows that feature selection stability in EntryPrune is comparable to NeuroFS and can be adjusted via the $c_{\mathrm{ratio}}$ parameter.

## D    ENTRYPRUNE FLEX

To address sensitivity to the $c_{\mathrm{ratio}}$ hyperparameter, we introduce EntryPrune flex, which dynamically adjusts the input layer size during training based on the behavior of the loss function. It extends Algorithm 1 between Steps 12 and 13, i.e., prior to selecting new candidate features, and is detailed in pseudocode in Algorithm 2.

---

**Algorithm 2** EntryPrune flex

1: **Initialize:** Loss before the last change of the network size $l_{\mathrm{change}}$, running loss $l$, and set the initial adjustment direction to shrinking.
2: **if** $l$ has not decreased for 10 rotations **then**
3:     **if** $l > l_{\mathrm{change}}$ **then**
4:         Change the direction: shrink $\leftrightarrow$ grow
5:     **end if**
6:     $l_{\mathrm{change}} = l$
7:     **if** direction is shrink **then**
8:         $c_{\mathrm{ratio}} = \max(\frac{1}{2} c_{\mathrm{ratio}}, \frac{1}{5} \frac{K}{N-K})$
9:     **else if** direction is grow **then**
10:         $c_{\mathrm{ratio}} = \min(2\, c_{\mathrm{ratio}}, 1)$
11:     **end if**
12: **end if**

---

**Key mechanism.**    EntryPrune flex monitors the running loss, $l$, and compares it with the loss recorded at the time of the last input layer size change, $l_{\mathrm{change}}$. If the loss stagnates (i.e., does not decrease for a fixed number of rotations), the algorithm adjusts the input layer size. Specifically:

1. Direction adjustment: If the loss increases compared to $l_{\mathrm{change}}$, the direction of change (shrink or grow) is reversed

2. Size adjustment: Depending on the direction, $c_{\mathrm{ratio}}$ is halved or doubled, bounded by predefined limits. The upper limit of $c_{\mathrm{ratio}} = 1$ represents using the maximum number of candidates, $N - K$, while the lower limit ensures a minimum input layer size of $\frac{6}{5} K$. These adjustment factors and bounds were found to work well in our preliminary experiments.

**Rationale.**    A well-balanced input layer size allows the network to explore a sufficient pool of candidate features in the presence of random regrowth. Shrinking the input layer promotes stability, while growing it enables exploration of additional candidates. The dynamic adjustment ensures that the network can escape suboptimal configurations.

**Practical considerations.**    The running loss $l$ as well as the loss at the time of input layer change, $l_{\mathrm{change}}$, can be either a training or validation loss, depending on whether the algorithm is used with a validation set. In our experiments, we use a validation set, which is detailed in Appendix E.2.

## E    EXPERIMENTAL SETUP

This appendix outlines the general experimental setup, including dataset characteristics, evaluation procedures, and model configurations used throughout our study. Each experimental configuration (i.e., a combination of dataset, feature selection method, and number of selected features) was run five

times, except for NeuroFS on the CIFAR datasets, where the number of runs was limited to a single iteration to ensure that runtime remained within feasible limits (below 12 hours per configuration).

The datasets and their dimensions and domains are summarized in Table 2. They provide a com-

Table 2: Dataset dimensions and domain

|  | Cases | Features | Domain | Reference |
|---|---|---|---|---|
| **Long Datasets** | | | | |
| CIFAR-10 | 60000 | 3072 | Image | Krizhevsky (2009) |
| CIFAR-100 | 60000 | 3072 | Image | Krizhevsky (2009) |
| COIL-20 | 1440 | 1024 | Image | Nene et al. (1996) |
| HAR | 10299 | 561 | Smartphone Sensor | Anguita et al. (2013) |
| ISOLET | 7797 | 617 | Speech | Fanty & Cole (1990) |
| MNIST | 70000 | 784 | Image | Deng (2012) |
| Fashion-MNIST | 70000 | 784 | Image | Xiao et al. (2017) |
| USPS | 9298 | 256 | Image | Hull (1994) |
| **Wide Datasets** | | | | |
| ARCENE | 200 | 10000 | Genomics | Guyon et al. (2004) |
| BASEHOCK | 1993 | 4862 | Text | Lang (1995) |
| GLA-BRA-180 | 180 | 49151 | Genomics | Sun et al. (2006) |
| Prostate-GE | 102 | 5966 | Genomics | Nie et al. (2010) |
| SMK | 187 | 19993 | Genomics | Spira et al. (2007) |

prehensive basis for comparison through their overlap with previous experiments (Yamada et al., 2020; Lemhadri et al., 2021; Liu et al., 2024). We categorize datasets as long if they have more cases than features, and vice versa as wide. The datasets all represent classification tasks and span different content domains, including speech processing (ISOLET), image recognition (MNIST), and smartphone sensor data (HAR). They are all freely available.

To ensure a fair comparison between embedded and filter methods, all experimental configurations include downstream learners. Initially, the data is split into training and test sets. Feature selection is performed using the training data, followed by training a downstream predictive model on the training data using only the selected features. The accuracy of the downstream learner is then evaluated on the test data. The number of selected features, $K$, is set to 25, 50, 75, and 100 in our experiments, following the values used by Atashgahi et al. (2023) and remaining consistent with ranges commonly adopted in related studies.[3] The downstream learners are classifiers based on a Support Vector Machine (SVM, Chang & Lin, 2011), K-Nearest Neighbors (KNN), and ExtraTrees (ET, Geurts et al., 2006). To align with the protocol of Atashgahi et al. (2023), the SVM classifier is used for all values of $K$, while KNN and ET are only used for $K = 50$. Experiments are conducted on an NVIDIA GeForce RTX 3060 GPU with 6GB of memory.

### E.1    BASELINES

We implemented the following baselines for comparison, in addition to a set of methods carried over from previous work, described below.

- XGBoost (Chen & Guestrin, 2016): A gradient boosting tree method that ranks features by their total loss reduction (gain) across all splits. We replicated Chen et al. (2020), using 500 boosting trees, a maximum depth of 15, and otherwise default parameters.

- SHAP-XGBoost (Lundberg et al., 2020): We pair the same XGBoost configuration described above with post-hoc SHAP value computation. SHAP values are calculated for all samples, and per-feature importance is obtained by averaging the absolute SHAP values across cases. A comparable use of SHAP for feature selection appears in Wang et al. (2024).

- CancelOut (Borisov et al., 2019): A method embedded in neural networks that applies trainable, regularized, sigmoid-scaled weights to input features. We followed the original

---

[3] Atashgahi et al. (2023) also used higher values for $K$ which are omitted in this study since there was little variance in the results between the different methods.

publication and code (https://github.com/unnir/CancelOut), using a three-hidden-layer architecture, the Adam optimizer with learning rate 0.003, $\beta_1 = 0.9$, $\beta_2 = 0.999$, and $\epsilon = 10^{-9}$, as well as early stopping. Hidden layer sizes were set to 10 plus the number of input features, but capped at 1000 for computational feasibility, with all other parameters kept as in the original code.

- GradEnFS (Liu et al., 2024): A method embedded in sparse neural networks that ranks input features based on the sensitivity to the loss. We used the hyperparameters from the original publication and code (https://github.com/KaitingLiu/GradEnFS), consisting of a three-hidden-layer network with 1000 neurons per layer. The choice of updating after each batch or each epoch was determined on a per-dataset basis according to which option gave better results in the original study, with batch updates used as the default otherwise.

Due to computational constraints, XGBoost, SHAP-XGBoost and CancelOut were not applied to the CIFAR datasets, and GradEnFS was not applied to MNIST or CIFAR.

The following baselines are carried over from Atashgahi et al. (2023), with implementation notes available in that work:

- Fisher Score (Gu et al., 2011): A classic filter method that selects feature sets based on their ability to separate data points.
- CIFE (Conditional Infomax Feature Extraction, Lin & Tang, 2006): A filter method that aims to maximize the class-relevant information of the feature set.
- ICAP (Interaction Capping Criterion, Jakulin, 2005): A filter method that considers the complementary relationship between features.
- RFS (Robust Feature Selection, Nie et al., 2010): A method embedded in regression that uses joint $L^1$ and $L^2$ regularization of the weights.
- QS (Quick Selection, Atashgahi et al., 2022): A method embedded in sparse neural networks that combines denoising autoencoders and the $L^1$ norm of first layer neuron weights.
- STG (Stochastic Gates, Yamada et al., 2020): A method embedded in neural networks that controls the input layer neurons using a trainable probabilistic gate.
- LassoNet (Lemhadri et al., 2021): A method embedded in neural networks that adds a regularized residual connection from the input layer to the output. The residual connection controls the sizes of first layer weights.
- RigL (Evci et al., 2020): A method embedded in sparse neural networks that rotates features by pruning based on parameter weights and regrowing based on gradients. Feature selection can be performed by investigating first layer weights after training (Atashgahi et al., 2023).
- NeuroFS (Atashgahi et al., 2023): A method embedded in sparse neural networks that extends the ideas used in RigL to input neurons.

Due to computational constraints, we conducted only a subset of baseline comparisons for the datasets added in our experiment (CIFAR-10, CIFAR-100, SMK, BASEHOCK).

### E.2 ENTRYPRUNE SETUP

The parameters used for EntryPrune in the main experiment are as follows. We employ a single hidden layer neural network with 100 neurons and a ReLU activation function, following the default architecture used in LassoNet. For training, we use a batch size of 1024 and a learning rate of 0.001 for the Adam optimizer. If there are fewer cases in the dataset, full batches are used instead. The hyperparameters specific to our method are: $c_{\mathrm{ratio}} = 0.2$ and $n_{\mathrm{mb}} = 100$ for long datasets, and $c_{\mathrm{ratio}} = 0.5$ and $n_{\mathrm{mb}} = 5$ for wide datasets.

Stopping is based on a combination of validation loss and the identified feature set. For this, the training data is split again into a training and a validation set. Training continues on the training set until the validation loss does not decrease for 100 input layer rotations or the set of $K$ features with the highest values in $e$ remains unchanged for 100 rotations. Afterwards, the training is again performed on the complete training data for the determined number of rotations. For the flex algorithm, during

this final training phase, the input layer is scaled from its initial size to the final size using a total of ten size change steps. We explore different stopping criteria and their hyperparameter settings in Appendix H, providing further insights into how they impact performance.

# F DETAILED RESULTS

Table 3: Resulting accuracy of the studied methods for different numbers of selected features $K$ and long datasets using the SVM downstream learner. Our proposed methods are "EntryPrune" and "EntryPrune flex". "All" is the accuracy using all features in the dataset. The best and second-best methods for each combination of $K$ and dataset are marked in bold and underlined, respectively. Entries represent the mean $\pm$ standard deviation of the downstream learner accuracy across five runs. Datasets marked with an asterisk were evaluated with a limited set of baseline methods (see Appendix E), while baseline results for the other datasets are reproduced from Atashgahi et al. (2023).

| | CIFAR-10* | CIFAR-100* | COIL-20 | HAR | ISOLET | MNIST | Fashion-MNIST | USPS |
|---|---|---|---|---|---|---|---|---|
| All | 54.36 | 26.39 | 100.00 | 95.05 | 96.03 | 97.92 | 88.30 | 97.58 |
| **K = 25** | | | | | | | | |
| XGBoost | - | - | 97.92 ± 0.00 | 90.67 ± 0.00 | 75.83 ± 0.00 | 69.07 ± 0.00 | 78.09 ± 0.00 | 92.96 ± 0.00 |
| SHAP-XGBoost | - | - | 95.49 ± 0.00 | 90.13 ± 0.00 | 87.12 ± 0.00 | 87.79 ± 0.00 | 74.12 ± 0.00 | 94.09 ± 0.00 |
| CancelOut | - | - | 74.51 ± 9.50 | 82.61 ± 1.13 | 66.33 ± 5.63 | 85.61 ± 1.16 | 76.16 ± 2.18 | 90.32 ± 1.45 |
| GradEnFS | - | - | 96.67 ± 1.75 | 88.23 ± 1.24 | 85.23 ± 3.81 | - | - | 93.16 ± 0.71 |
| NeuroFS | 40.40 | **17.40** | 95.86 ± 1.31 | 87.46 ± 0.79 | 86.22 ± 0.84 | 87.86 ± 1.77 | 79.38 ± 0.96 | 93.98 ± 0.87 |
| LassoNet | 23.30 ± 0.96 | 9.58 ± 1.05 | 92.72 ± 0.85 | **93.00 ± 0.31** | 76.48 ± 0.39 | 86.40 ± 1.26 | 78.68 ± 0.55 | 94.04 ± 0.38 |
| STG | 36.88 ± 0.59 | 11.74 ± 0.76 | 97.02 ± 1.41 | 87.48 ± 0.80 | 77.16 ± 4.34 | 85.24 ± 1.89 | 77.44 ± 0.53 | 94.04 ± 0.46 |
| QS | - | - | 91.00 ± 4.21 | 87.14 ± 1.74 | 72.56 ± 6.53 | 85.25 ± 1.47 | 71.57 ± 1.97 | 93.00 ± 0.81 |
| Fisher | 18.55 ± 0.00 | 4.60 ± 0.00 | 24.70 ± 0.00 | 77.10 ± 0.00 | 57.40 ± 0.00 | 74.40 ± 0.00 | 53.10 ± 0.00 | 82.00 ± 0.00 |
| CIFE | - | - | 50.70 ± 0.00 | 80.20 ± 0.00 | 56.00 ± 0.00 | 80.90 ± 0.00 | 63.40 ± 0.00 | 50.20 ± 0.00 |
| ICAP | - | - | 94.40 ± 0.00 | 84.50 ± 0.00 | 67.10 ± 0.00 | 81.60 ± 0.00 | 50.10 ± 0.00 | 89.90 ± 0.00 |
| RFS | - | - | 88.20 ± 0.00 | 88.90 ± 0.00 | 76.50 ± 0.00 | | | 94.80 ± 0.00 |
| RigL | - | - | 92.38 ± 3.20 | 86.46 ± 1.47 | 79.98 ± 2.25 | 82.06 ± 0.99 | 74.12 ± 1.59 | 93.10 ± 0.62 |
| EntryPrune | 40.71 ± 1.75 | 15.33 ± 0.84 | 98.75 ± 0.31 | 92.07 ± 1.42 | **88.45 ± 1.16** | 93.04 ± 0.41 | **83.05 ± 0.40** | **95.82 ± 0.49** |
| EntryPrune flex | **41.82 ± 0.64** | 15.26 ± 1.09 | **98.89 ± 0.71** | 92.06 ± 0.97 | 88.28 ± 1.41 | **93.10 ± 0.25** | 82.70 ± 0.32 | 95.68 ± 0.08 |
| **K = 50** | | | | | | | | |
| XGBoost | - | - | 98.61 ± 0.00 | 90.36 ± 0.00 | 88.72 ± 0.00 | 89.04 ± 0.00 | 83.30 ± 0.00 | 95.54 ± 0.00 |
| SHAP-XGBoost | - | - | 99.31 ± 0.00 | 92.94 ± 0.00 | 92.95 ± 0.00 | 93.54 ± 0.00 | 77.91 ± 0.00 | 95.54 ± 0.00 |
| CancelOut | - | - | 89.24 ± 3.38 | 90.60 ± 1.01 | 80.85 ± 3.99 | 93.15 ± 0.60 | 83.33 ± 0.45 | 94.90 ± 0.49 |
| GradEnFS | - | - | 98.68 ± 0.29 | 91.13 ± 1.56 | - | - | - | 96.26 ± 0.51 |
| NeuroFS | 46.30 | **21.10** | 98.78 ± 0.29 | 91.46 ± 0.72 | 92.62 ± 0.40 | 95.30 ± 0.41 | 83.78 ± 0.64 | 96.78 ± 0.17 |
| LassoNet | 28.77 ± 5.00 | 10.55 ± 0.50 | 97.16 ± 1.06 | 93.74 ± 0.39 | 84.90 ± 0.22 | 94.46 ± 0.21 | 82.58 ± 0.10 | 95.94 ± 0.15 |
| STG | 42.70 ± 0.42 | 14.08 ± 1.25 | 99.32 ± 0.40 | 91.22 ± 1.23 | 85.82 ± 2.83 | 93.20 ± 0.62 | 82.36 ± 0.52 | 96.62 ± 0.34 |
| QS | - | - | 96.52 ± 1.53 | 91.96 ± 1.04 | 89.78 ± 1.80 | 93.62 ± 0.49 | 80.82 ± 0.51 | 95.52 ± 0.27 |
| Fisher | 20.52 ± 0.00 | 5.21 ± 0.00 | 74.00 ± 0.00 | 79.80 ± 0.00 | 67.40 ± 0.00 | 81.90 ± 0.00 | 67.80 ± 0.00 | 91.00 ± 0.00 |
| CIFE | - | - | 59.40 ± 0.00 | 84.20 ± 0.00 | 59.80 ± 0.00 | 89.30 ± 0.00 | 66.90 ± 0.00 | 61.30 ± 0.00 |
| ICAP | - | - | 99.30 ± 0.00 | 88.70 ± 0.00 | 75.10 ± 0.00 | 89.00 ± 0.00 | 59.50 ± 0.00 | 95.20 ± 0.00 |
| RFS | - | - | 95.80 ± 0.00 | **94.00 ± 0.00** | 91.50 ± 0.00 | | | 95.80 ± 0.00 |
| RigL | - | - | 97.86 ± 1.32 | 91.82 ± 0.30 | 89.58 ± 1.24 | 93.94 ± 0.63 | 81.92 ± 0.87 | 96.04 ± 0.58 |
| EntryPrune | 46.65 ± 0.60 | 18.98 ± 0.90 | **99.58 ± 0.29** | 93.74 ± 0.62 | 93.41 ± 0.25 | 96.69 ± 0.19 | **85.95 ± 0.22** | 96.83 ± 0.17 |
| EntryPrune flex | **47.23 ± 0.87** | 19.13 ± 1.34 | 99.51 ± 0.19 | 93.65 ± 0.36 | **93.46 ± 0.19** | **96.79 ± 0.11** | 85.84 ± 0.36 | **97.06 ± 0.23** |
| **K = 75** | | | | | | | | |
| XGBoost | - | - | **100.00 ± 0.00** | 91.11 ± 0.00 | 93.46 ± 0.00 | 93.33 ± 0.00 | 84.40 ± 0.00 | 96.18 ± 0.00 |
| SHAP-XGBoost | - | - | 99.65 ± 0.00 | 94.16 ± 0.00 | 94.42 ± 0.00 | 95.64 ± 0.00 | 83.15 ± 0.00 | 96.08 ± 0.00 |
| CancelOut | - | - | 96.39 ± 2.66 | 92.87 ± 1.06 | 89.58 ± 1.75 | 95.32 ± 0.26 | 85.37 ± 0.21 | 96.17 ± 0.24 |
| GradEnFS | - | - | 99.44 ± 0.40 | 94.35 ± 0.59 | 94.17 ± 0.30 | - | - | 96.84 ± 0.21 |
| NeuroFS | **50.30** | **22.10** | 99.06 ± 0.12 | 93.16 ± 0.79 | 94.04 ± 0.34 | 96.76 ± 0.22 | 85.70 ± 0.28 | 97.06 ± 0.15 |
| LassoNet | 30.22 ± 1.54 | 12.41 ± 2.15 | 99.46 ± 0.35 | 94.62 ± 0.17 | 91.00 ± 0.62 | 96.00 ± 0.09 | 83.92 ± 0.13 | 96.36 ± 0.08 |
| STG | 46.20 ± 0.72 | 16.21 ± 0.60 | 99.68 ± 0.22 | 92.42 ± 1.11 | 90.10 ± 2.17 | 95.52 ± 0.22 | 84.14 ± 0.43 | 96.88 ± 0.23 |
| QS | - | - | 98.17 ± 1.16 | 93.50 ± 0.77 | 93.04 ± 0.46 | 95.98 ± 0.33 | 83.80 ± 0.53 | 96.85 ± 0.05 |
| Fisher | 22.08 ± 0.00 | 6.05 ± 0.00 | 76.00 ± 0.00 | 81.70 ± 0.00 | 76.00 ± 0.00 | 87.10 ± 0.00 | 74.30 ± 0.00 | 94.40 ± 0.00 |
| CIFE | - | - | 63.20 ± 0.00 | 84.80 ± 0.00 | 74.30 ± 0.00 | 92.70 ± 0.00 | 67.70 ± 0.00 | 68.00 ± 0.00 |
| ICAP | - | - | 99.00 ± 0.00 | 89.20 ± 0.00 | 79.70 ± 0.00 | 92.40 ± 0.00 | 67.20 ± 0.00 | 95.30 ± 0.00 |
| RFS | - | - | 99.70 ± 0.00 | 94.90 ± 0.00 | 93.90 ± 0.00 | | | 97.20 ± 0.00 |
| RigL | - | - | 99.20 ± 0.43 | 93.34 ± 0.47 | 92.32 ± 0.56 | 95.98 ± 0.51 | 84.52 ± 0.72 | 96.90 ± 0.24 |
| EntryPrune | 49.28 ± 0.47 | 20.40 ± 0.63 | **99.93 ± 0.16** | **95.31 ± 0.37** | 94.60 ± 0.49 | 97.49 ± 0.13 | 86.75 ± 0.25 | 97.15 ± 0.19 |
| EntryPrune flex | 49.65 ± 0.38 | 21.52 ± 0.58 | **99.93 ± 0.16** | 94.60 ± 0.65 | **94.88 ± 0.31** | **97.53 ± 0.11** | **86.76 ± 0.14** | **97.19 ± 0.10** |
| **K = 100** | | | | | | | | |
| XGBoost | - | - | **100.00 ± 0.00** | 91.89 ± 0.00 | 93.65 ± 0.00 | 95.60 ± 0.00 | 85.50 ± 0.00 | 96.40 ± 0.00 |
| SHAP-XGBoost | - | - | 99.65 ± 0.00 | 94.60 ± 0.00 | 94.81 ± 0.00 | 96.58 ± 0.00 | 85.03 ± 0.00 | 96.56 ± 0.00 |
| CancelOut | - | - | 98.06 ± 0.72 | 94.32 ± 0.43 | 91.90 ± 1.86 | 96.27 ± 0.28 | 86.51 ± 0.04 | 96.60 ± 0.15 |
| GradEnFS | - | - | 99.79 ± 0.31 | 94.61 ± 0.59 | 94.82 ± 0.35 | - | - | 97.19 ± 0.25 |
| NeuroFS | 49.50 | **23.20** | 99.18 ± 0.50 | 94.18 ± 0.29 | 95.06 ± 0.31 | 97.32 ± 0.17 | 86.64 ± 0.21 | 97.22 ± 0.12 |
| LassoNet | 32.12 ± 0.56 | 13.25 ± 2.22 | 99.30 ± 0.00 | 95.14 ± 0.29 | 93.18 ± 0.22 | 96.64 ± 0.14 | 84.98 ± 0.18 | 97.04 ± 0.12 |
| STG | 47.42 ± 0.40 | 18.34 ± 0.63 | 99.76 ± 0.12 | 92.82 ± 0.74 | 92.64 ± 0.56 | 96.38 ± 0.35 | 85.20 ± 0.58 | 97.08 ± 0.18 |
| QS | - | - | 98.28 ± 1.15 | 94.06 ± 0.48 | 94.22 ± 0.28 | 96.85 ± 0.09 | 85.52 ± 0.15 | 97.00 ± 0.14 |
| Fisher | 23.72 ± 0.00 | 6.62 ± 0.00 | 80.20 ± 0.00 | 83.80 ± 0.00 | 79.80 ± 0.00 | 90.70 ± 0.00 | 79.60 ± 0.00 | 96.50 ± 0.00 |
| CIFE | - | - | 67.70 ± 0.00 | 85.30 ± 0.00 | 81.20 ± 0.00 | 95.10 ± 0.00 | 69.20 ± 0.00 | 78.00 ± 0.00 |
| ICAP | - | - | **100.00 ± 0.00** | 92.10 ± 0.00 | 82.80 ± 0.00 | 95.00 ± 0.00 | 77.70 ± 0.00 | 95.40 ± 0.00 |
| RFS | - | - | **100.00 ± 0.00** | 95.40 ± 0.00 | 94.40 ± 0.00 | | | 97.40 ± 0.00 |
| RigL | - | - | 99.40 ± 0.43 | 94.08 ± 0.26 | 93.66 ± 0.58 | 96.88 ± 0.22 | 85.82 ± 0.23 | 97.14 ± 0.10 |
| EntryPrune | 50.58 ± 0.40 | 22.34 ± 0.43 | 99.93 ± 0.16 | **95.61 ± 0.25** | **95.73 ± 0.46** | **97.80 ± 0.10** | **87.32 ± 0.15** | 97.34 ± 0.15 |
| EntryPrune flex | **50.73 ± 0.34** | 23.19 ± 0.18 | **100.00 ± 0.00** | 95.19 ± 0.19 | 95.21 ± 0.23 | 97.79 ± 0.07 | 87.21 ± 0.08 | 97.37 ± 0.13 |

Table 4: Resulting accuracy of the studied methods for different numbers of selected features $K$ and wide datasets using the SVM downstream learner. Our proposed methods are "EntryPrune" and "EntryPrune flex". "All" is the accuracy using all features in the dataset. The best and second-best methods for each combination of $K$ and dataset are marked in bold and underlined, respectively. Entries represent the mean $\pm$ standard deviation of the downstream learner accuracy across five runs. Datasets marked with an asterisk were evaluated with a limited set of baseline methods (see Appendix E), while baseline results for the other datasets are reproduced from Atashgahi et al. (2023).

| | ARCENE | BASEHOCK* | GLA-BRA-180 | Prostate-GE | SMK* |
|---|---|---|---|---|---|
| All | 77.50 | 94.24 | 72.22 | 80.95 | 84.21 |
| **K = 25** | | | | | |
| XGBoost | 77.50 ± 0.00 | 85.21 ± 0.00 | 75.00 ± 0.00 | 90.48 ± 0.00 | 68.42 ± 0.00 |
| SHAP-XGBoost | 72.50 ± 0.00 | 86.47 ± 0.00 | **80.56 ± 0.00** | 90.48 ± 0.00 | 71.05 ± 0.00 |
| CancelOut | 63.50 ± 9.45 | 85.36 ± 1.95 | 56.11 ± 7.71 | **92.38 ± 2.61** | 75.26 ± 3.99 |
| GradEnFS | 63.50 ± 6.02 | 88.37 ± 1.20 | 65.00 ± 4.21 | 87.62 ± 2.61 | _77.89 ± 3.53_ |
| NeuroFS | 63.00 ± 4.85 | 85.46 ± 2.10 | 73.88 ± 3.80 | 88.58 ± 2.35 | 77.34 ± 5.76 |
| LassoNet | 69.00 ± 2.55 | **89.82 ± 1.21** | 76.12 ± 4.19 | 88.58 ± 2.35 | 74.74 ± 3.00 |
| STG | 69.00 ± 5.15 | 78.90 ± 3.17 | 67.22 ± 4.78 | 85.72 ± 3.00 | 76.84 ± 5.06 |
| QS | 73.75 ± 8.20 | - | 69.45 ± 2.75 | 71.43 ± 12.16 | - |
| Fisher | 65.00 ± 0.00 | 54.64 ± 0.00 | 58.30 ± 0.00 | _90.50 ± 0.00_ | 76.32 ± 0.00 |
| CIFE | 67.50 ± 0.00 | - | 61.10 ± 0.00 | 61.90 ± 0.00 | - |
| ICAP | 77.50 ± 0.00 | - | 69.40 ± 0.00 | 47.60 ± 0.00 | - |
| RFS | 77.50 ± 0.00 | - | - | 90.50 ± 0.00 | - |
| RigL | 74.50 ± 4.30 | - | 66.10 ± 3.22 | 78.08 ± 6.46 | - |
| EntryPrune | _78.50 ± 6.52_ | 82.31 ± 1.82 | 75.00 ± 2.78 | 90.48 ± 0.00 | **82.63 ± 6.06** |
| EntryPrune flex | **80.50 ± 5.12** | 81.40 ± 0.88 | _77.78 ± 1.96_ | 88.57 ± 2.61 | 77.89 ± 6.34 |
| **K = 50** | | | | | |
| XGBoost | 72.50 ± 0.00 | 90.48 ± 0.00 | 72.22 ± 0.00 | 90.48 ± 0.00 | 76.32 ± 0.00 |
| SHAP-XGBoost | **77.50 ± 0.00** | 89.47 ± 0.00 | _77.78 ± 0.00_ | _90.48 ± 0.00_ | 76.32 ± 0.00 |
| CancelOut | 58.50 ± 2.85 | 81.70 ± 5.60 | 58.89 ± 6.63 | 87.62 ± 4.26 | 78.42 ± 1.18 |
| GradEnFS | 69.00 ± 7.83 | _91.38 ± 0.91_ | 64.44 ± 6.63 | 87.62 ± 4.26 | 77.37 ± 2.35 |
| NeuroFS | 76.50 ± 2.55 | 88.08 ± 0.70 | **80.54 ± 4.96** | **90.50 ± 0.00** | 81.56 ± 2.65 |
| LassoNet | 71.00 ± 2.00 | **91.98 ± 1.16** | 74.46 ± 4.78 | 88.58 ± 2.35 | 80.53 ± 3.99 |
| STG | 71.00 ± 2.55 | 84.41 ± 3.20 | 70.00 ± 4.08 | 84.78 ± 3.55 | 81.58 ± 1.86 |
| QS | 74.38 ± 4.80 | - | 72.20 ± 2.80 | 76.20 ± 7.53 | - |
| Fisher | 67.50 ± 0.00 | 62.16 ± 0.00 | 63.90 ± 0.00 | **90.50 ± 0.00** | 78.95 ± 0.00 |
| CIFE | 52.50 ± 0.00 | - | 58.30 ± 0.00 | 47.60 ± 0.00 | - |
| ICAP | 70.00 ± 0.00 | - | 72.20 ± 0.00 | 57.10 ± 0.00 | - |
| RFS | **77.50 ± 0.00** | - | - | **90.50 ± 0.00** | - |
| RigL | _77.00 ± 3.32_ | - | 70.54 ± 4.16 | 79.06 ± 7.11 | - |
| EntryPrune | 72.50 ± 5.59 | 86.47 ± 1.45 | 73.33 ± 1.52 | _90.48 ± 0.00_ | **83.68 ± 4.32** |
| EntryPrune flex | 76.00 ± 6.75 | 84.56 ± 1.67 | 74.44 ± 2.32 | 89.52 ± 2.13 | _82.11 ± 3.43_ |
| **K = 75** | | | | | |
| XGBoost | 77.50 ± 0.00 | 90.48 ± 0.00 | 72.22 ± 0.00 | 90.48 ± 0.00 | 73.68 ± 0.00 |
| SHAP-XGBoost | 75.00 ± 0.00 | _91.73 ± 0.00_ | _77.78 ± 0.00_ | _90.48 ± 0.00_ | 76.32 ± 0.00 |
| CancelOut | 61.00 ± 8.02 | 88.37 ± 3.16 | 65.56 ± 7.51 | 88.57 ± 4.26 | 80.00 ± 5.13 |
| GradEnFS | 71.50 ± 4.87 | 91.53 ± 0.65 | 68.89 ± 7.45 | 87.62 ± 4.26 | 81.05 ± 4.32 |
| NeuroFS | **82.00 ± 4.00** | 90.86 ± 2.20 | **82.24 ± 3.31** | 89.54 ± 1.92 | 78.40 ± 3.89 |
| LassoNet | 70.50 ± 2.45 | **91.88 ± 1.01** | 76.64 ± 5.44 | **90.50 ± 0.00** | 78.42 ± 7.54 |
| STG | 75.00 ± 2.74 | 86.42 ± 3.36 | 71.08 ± 1.37 | 84.78 ± 3.55 | 81.58 ± 3.22 |
| QS | 76.88 ± 2.72 | - | 73.60 ± 1.40 | 72.62 ± 9.78 | - |
| Fisher | 70.00 ± 0.00 | 65.16 ± 0.00 | 66.70 ± 0.00 | **90.50 ± 0.00** | **84.21 ± 0.00** |
| CIFE | 72.50 ± 0.00 | - | 58.30 ± 0.00 | 47.60 ± 0.00 | - |
| ICAP | 72.50 ± 0.00 | - | 72.20 ± 0.00 | 57.10 ± 0.00 | - |
| RFS | 80.00 ± 0.00 | - | - | **90.50 ± 0.00** | - |
| RigL | _81.50 ± 4.64_ | - | 72.22 ± 4.98 | 79.06 ± 8.83 | - |
| EntryPrune | 71.00 ± 7.42 | 87.47 ± 1.59 | _77.78 ± 3.40_ | _90.48 ± 0.00_ | 82.63 ± 2.35 |
| EntryPrune flex | **82.00 ± 4.81** | 86.87 ± 1.69 | 75.56 ± 3.04 | _90.48 ± 0.00_ | _83.16 ± 3.53_ |
| **K = 100** | | | | | |
| XGBoost | 80.00 ± 0.00 | _92.73 ± 0.00_ | 75.00 ± 0.00 | _90.48 ± 0.00_ | 76.32 ± 0.00 |
| SHAP-XGBoost | 72.50 ± 0.00 | **93.48 ± 0.00** | 75.00 ± 0.00 | _90.48 ± 0.00_ | 76.32 ± 0.00 |
| CancelOut | 61.50 ± 4.18 | 89.42 ± 1.11 | 67.78 ± 7.24 | 89.52 ± 2.13 | 81.05 ± 4.71 |
| GradEnFS | 72.00 ± 5.42 | 92.53 ± 1.45 | 70.56 ± 8.91 | 89.52 ± 2.13 | 80.00 ± 6.34 |
| NeuroFS | _82.00 ± 1.87_ | 91.62 ± 2.08 | **81.12 ± 2.05** | 89.54 ± 1.92 | 79.48 ± 5.69 |
| LassoNet | 72.00 ± 4.30 | 92.08 ± 0.52 | _79.46 ± 2.83_ | **90.50 ± 0.00** | 78.42 ± 2.20 |
| STG | 75.50 ± 3.67 | 86.67 ± 1.66 | 72.20 ± 3.07 | 85.72 ± 3.00 | 82.63 ± 3.99 |
| QS | 78.12 ± 1.08 | - | 73.60 ± 1.40 | 78.58 ± 9.82 | - |
| Fisher | 65.00 ± 0.00 | 64.91 ± 0.00 | 66.70 ± 0.00 | **90.50 ± 0.00** | _84.21 ± 0.00_ |
| CIFE | 65.00 ± 0.00 | - | 58.30 ± 0.00 | 71.40 ± 0.00 | - |
| ICAP | **82.50 ± 0.00** | - | 69.40 ± 0.00 | 52.40 ± 0.00 | - |
| RFS | 80.00 ± 0.00 | - | - | **90.50 ± 0.00** | - |
| RigL | 80.00 ± 4.47 | - | 73.90 ± 3.76 | 81.92 ± 8.18 | - |
| EntryPrune | 74.00 ± 2.85 | 87.22 ± 1.42 | 77.22 ± 3.62 | _90.48 ± 0.00_ | 82.11 ± 2.88 |
| EntryPrune flex | 77.50 ± 3.06 | 87.72 ± 2.56 | 77.78 ± 4.39 | _90.48 ± 0.00_ | **85.26 ± 1.44** |

Table 5: Resulting accuracy of the studied methods for different downstream learners and long datasets using $K = 50$ selected features. Our proposed methods are "EntryPrune" and "EntryPrune flex". "All" is the accuracy using all features in the dataset. The best and second-best methods for each combination of learner and dataset are marked in bold and underlined, respectively. Entries represent the mean $\pm$ standard deviation of the downstream learner accuracy across five runs. Datasets marked with an asterisk were evaluated with a limited set of baseline methods (see Appendix E), while baseline results for the other datasets are reproduced from Atashgahi et al. (2023).

| | CIFAR-10* | CIFAR-100* | COIL-20 | HAR | ISOLET | MNIST | Fashion-MNIST | USPS |
|---|---|---|---|---|---|---|---|---|
| **Learner: ET** | | | | | | | | |
| All | 45.49 ± 0.23 | 20.77 ± 0.17 | 100.00 ± 0.00 | 93.53 ± 0.15 | 94.05 ± 0.32 | 97.10 ± 0.05 | 87.19 ± 0.13 | 96.29 ± 0.16 |
| XGBoost | - | - | 99.93 ± 0.16 | 90.72 ± 0.28 | 88.14 ± 0.59 | 89.12 ± 0.10 | 84.14 ± 0.16 | 94.06 ± 0.39 |
| SHAP-XGBoost | - | - | 99.86 ± 0.31 | 90.27 ± 0.10 | 91.85 ± 0.43 | 92.18 ± 0.14 | 79.53 ± 0.17 | 94.77 ± 0.15 |
| CancelOut | - | - | 97.99 ± 0.83 | 86.00 ± 0.83 | 78.88 ± 3.96 | 91.66 ± 0.67 | 83.56 ± 0.37 | 93.28 ± 0.51 |
| GradEnFS | - | - | 99.86 ± 0.19 | 87.45 ± 1.65 | 89.06 ± 1.44 | | | 94.92 ± 0.31 |
| NeuroFS | 39.70 | **17.30** | 99.94 ± 0.12 | 85.48 ± 1.46 | 91.46 ± 0.73 | 93.68 ± 0.43 | 84.26 ± 0.55 | 95.44 ± 0.27 |
| LassoNet | 28.05 ± 3.60 | 9.53 ± 0.37 | 99.76 ± 0.12 | **91.12 ± 0.30** | 84.94 ± 0.62 | 92.96 ± 0.15 | 83.68 ± 0.13 | 94.86 ± 0.22 |
| STG | 37.75 ± 0.35 | 12.77 ± 1.11 | 100.00 ± 0.00 | 88.68 ± 0.42 | 88.50 ± 2.15 | 90.38 ± 0.42 | 82.05 ± 0.48 | 94.32 ± 0.21 |
| QS | - | - | 99.25 ± 0.47 | 87.86 ± 0.72 | 88.78 ± 1.86 | 91.95 ± 0.58 | 81.28 ± 0.54 | 94.28 ± 0.40 |
| Fisher | 22.03 ± 0.13 | 5.87 ± 0.17 | 96.86 ± 0.43 | 85.50 ± 0.30 | 81.42 ± 0.59 | 84.86 ± 0.15 | 72.06 ± 0.08 | 90.94 ± 0.24 |
| CIFE | - | - | 74.70 ± 0.00 | 85.30 ± 0.00 | 55.40 ± 0.00 | 87.60 ± 0.00 | 68.40 ± 0.00 | 82.70 ± 0.00 |
| ICAP | - | - | 99.70 ± 0.00 | 89.20 ± 0.00 | 70.60 ± 0.00 | 87.80 ± 0.00 | 65.50 ± 0.00 | 93.50 ± 0.00 |
| RFS | - | - | 98.30 ± 0.00 | 89.70 ± 0.00 | 90.40 ± 0.00 | - | - | 94.70 ± 0.00 |
| EntryPrune | 40.73 ± 0.26 | 16.52 ± 0.48 | **100.00 ± 0.00** | 90.32 ± 1.26 | **92.65 ± 0.52** | 95.30 ± 0.12 | **85.70 ± 0.22** | 95.76 ± 0.13 |
| EntryPrune flex | **40.99 ± 0.55** | 16.60 ± 0.48 | **100.00 ± 0.00** | 91.12 ± 1.33 | 92.19 ± 0.47 | **95.41 ± 0.21** | 85.49 ± 0.29 | **95.91 ± 0.18** |
| **Learner: KNN** | | | | | | | | |
| All | 35.39 | 17.55 | 100.00 | 87.85 | 88.14 | 96.91 | 84.96 | 97.37 |
| XGBoost | - | - | **100.00 ± 0.00** | 87.85 ± 0.00 | 82.05 ± 0.00 | 83.58 ± 0.00 | 80.41 ± 0.00 | 94.78 ± 0.00 |
| SHAP-XGBoost | - | - | **100.00 ± 0.00** | 89.62 ± 0.00 | **88.40 ± 0.00** | 90.29 ± 0.00 | 73.99 ± 0.00 | 95.05 ± 0.00 |
| CancelOut | - | - | 98.61 ± 0.89 | 84.51 ± 0.89 | 68.15 ± 4.05 | 88.83 ± 0.99 | 79.15 ± 0.47 | 93.74 ± 0.51 |
| GradEnFS | - | - | **100.00 ± 0.00** | 85.15 ± 1.55 | 83.26 ± 1.70 | | | 95.52 ± 0.38 |
| NeuroFS | 32.80 | **15.30** | 99.80 ± 0.28 | 84.64 ± 1.77 | 85.96 ± 1.53 | 91.64 ± 0.57 | 80.12 ± 0.87 | 96.18 ± 0.49 |
| LassoNet | 21.18 ± 2.78 | 7.31 ± 0.30 | 98.84 ± 0.20 | 88.70 ± 0.57 | 79.22 ± 0.47 | 91.38 ± 0.36 | 79.30 ± 0.20 | 95.70 ± 0.26 |
| STG | 30.79 ± 0.60 | 10.40 ± 0.92 | 99.94 ± 0.12 | 87.86 ± 0.39 | 83.16 ± 3.42 | 87.16 ± 0.64 | 77.65 ± 0.48 | 95.14 ± 0.45 |
| QS | - | - | 98.80 ± 0.38 | 85.88 ± 1.13 | 82.38 ± 3.12 | 89.30 ± 0.76 | 76.65 ± 0.51 | 95.17 ± 0.45 |
| Fisher | 17.01 ± 0.00 | 4.89 ± 0.00 | 95.80 ± 0.00 | 81.10 ± 0.00 | 74.10 ± 0.00 | 80.20 ± 0.00 | 63.70 ± 0.00 | 88.80 ± 0.00 |
| CIFE | - | - | 71.20 ± 0.00 | 71.80 ± 0.00 | 44.60 ± 0.00 | 82.90 ± 0.00 | 61.60 ± 0.00 | 59.60 ± 0.00 |
| ICAP | - | - | 98.60 ± 0.00 | 82.70 ± 0.00 | 59.00 ± 0.00 | 83.40 ± 0.00 | 59.30 ± 0.00 | 94.00 ± 0.00 |
| RFS | - | - | 97.20 ± 0.00 | **90.30 ± 0.00** | 87.20 ± 0.00 | - | - | 95.40 ± 0.00 |
| EntryPrune | 32.70 ± 0.68 | 13.86 ± 0.56 | 99.93 ± 0.16 | 86.43 ± 0.93 | 88.21 ± 0.46 | 94.48 ± 0.20 | 82.01 ± 0.20 | **96.65 ± 0.20** |
| EntryPrune flex | **33.21 ± 0.71** | **13.98 ± 0.52** | 99.79 ± 0.31 | 86.40 ± 1.14 | 87.12 ± 0.69 | **94.62 ± 0.24** | 82.10 ± 0.56 | 96.48 ± 0.39 |
| **Learner: SVM** | | | | | | | | |
| All | 54.36 | 26.39 | 100.00 | 95.05 | 96.03 | 97.92 | 88.30 | 97.58 |
| XGBoost | - | - | 98.61 ± 0.00 | 90.36 ± 0.00 | 88.72 ± 0.00 | 89.04 ± 0.00 | 83.30 ± 0.00 | 95.54 ± 0.00 |
| SHAP-XGBoost | - | - | 99.31 ± 0.00 | 92.94 ± 0.00 | 92.95 ± 0.00 | 93.54 ± 0.00 | 77.91 ± 0.00 | 95.54 ± 0.00 |
| CancelOut | - | - | 89.24 ± 3.38 | 90.60 ± 1.01 | 80.85 ± 3.99 | 93.15 ± 0.60 | 83.33 ± 0.45 | 94.90 ± 0.49 |
| GradEnFS | - | - | 98.68 ± 0.29 | 92.68 ± 1.24 | 91.13 ± 1.56 | | | 96.26 ± 0.51 |
| NeuroFS | 46.30 | **21.10** | 98.78 ± 0.29 | 91.46 ± 0.72 | 92.62 ± 0.40 | 95.30 ± 0.41 | 83.78 ± 0.64 | 96.78 ± 0.17 |
| LassoNet | 28.77 ± 5.00 | 10.55 ± 0.50 | 97.16 ± 1.06 | 93.74 ± 0.39 | 84.90 ± 0.22 | 94.46 ± 0.21 | 82.58 ± 0.10 | 95.94 ± 0.15 |
| STG | 42.70 ± 0.42 | 14.08 ± 1.25 | 99.32 ± 0.40 | 91.22 ± 1.23 | 85.82 ± 2.83 | 93.20 ± 0.62 | 82.36 ± 0.52 | 96.62 ± 0.34 |
| QS | - | - | 96.52 ± 1.53 | 91.96 ± 1.04 | 89.78 ± 1.80 | 93.62 ± 0.49 | 80.82 ± 0.51 | 95.52 ± 0.27 |
| Fisher | 20.52 ± 0.00 | 5.21 ± 0.00 | 74.00 ± 0.00 | 79.80 ± 0.00 | 67.40 ± 0.00 | 81.90 ± 0.00 | 67.80 ± 0.00 | 91.00 ± 0.00 |
| CIFE | - | - | 59.40 ± 0.00 | 84.20 ± 0.00 | 59.80 ± 0.00 | 89.30 ± 0.00 | 66.90 ± 0.00 | 61.30 ± 0.00 |
| ICAP | - | - | 99.30 ± 0.00 | 88.70 ± 0.00 | 75.10 ± 0.00 | 89.00 ± 0.00 | 59.50 ± 0.00 | 95.20 ± 0.00 |
| RFS | - | - | 95.80 ± 0.00 | 91.50 ± 0.00 | **94.00 ± 0.00** | - | - | 95.80 ± 0.00 |
| RigL | - | - | 97.86 ± 1.32 | 91.82 ± 0.30 | 89.58 ± 1.24 | 93.94 ± 0.63 | 81.92 ± 0.87 | 96.04 ± 0.58 |
| EntryPrune | 46.65 ± 0.60 | 18.98 ± 0.90 | **99.58 ± 0.29** | **93.74 ± 0.62** | 93.41 ± 0.25 | 96.69 ± 0.19 | **85.95 ± 0.22** | 96.83 ± 0.17 |
| EntryPrune flex | **47.23 ± 0.87** | 19.13 ± 1.34 | 99.51 ± 0.19 | 93.65 ± 0.36 | **93.46 ± 0.19** | **96.79 ± 0.11** | 85.84 ± 0.36 | **97.06 ± 0.23** |

Table 6: Resulting accuracy of the studied methods for different downstream learners and wide datasets using $K = 50$ selected features. Our proposed methods are "EntryPrune" and "EntryPrune flex". "All" is the accuracy using all features in the dataset. The best and second-best methods for each combination of learner and dataset are marked in bold and underlined, respectively. Entries represent the mean $\pm$ standard deviation of the downstream learner accuracy across five runs. Datasets marked with an asterisk were evaluated with a limited set of baseline methods (see Appendix E), while baseline results for the other datasets are reproduced from Atashgahi et al. (2023).

| | ARCENE | BASEHOCK* | GLA-BRA-180 | Prostate-GE | SMK* |
|---|---|---|---|---|---|
| **Learner: ET** | | | | | |
| All | $79.50 \pm 4.85$ | $97.09 \pm 0.34$ | $75.00 \pm 4.97$ | $88.57 \pm 3.81$ | $80.00 \pm 5.16$ |
| XGBoost | $77.00 \pm 2.09$ | $\mathbf{95.04 \pm 0.11}$ | $74.44 \pm 1.24$ | $\underline{90.48 \pm 0.00}$ | $76.32 \pm 3.22$ |
| SHAP-XGBoost | $75.00 \pm 0.00$ | $\underline{94.44 \pm 0.33}$ | $73.33 \pm 2.48$ | $\underline{90.48 \pm 0.00}$ | $74.74 \pm 2.35$ |
| CancelOut | $63.50 \pm 5.18$ | $85.51 \pm 6.54$ | $58.33 \pm 6.80$ | $90.48 \pm 3.37$ | $76.32 \pm 4.92$ |
| GradEnFS | $66.00 \pm 4.18$ | $93.63 \pm 2.01$ | $67.22 \pm 2.32$ | $85.71 \pm 3.37$ | $74.74 \pm 4.40$ |
| NeuroFS | $75.00 \pm 5.24$ | $90.44 \pm 1.86$ | $75.46 \pm 6.71$ | $\mathbf{90.50 \pm 0.00}$ | $78.96 \pm 4.55$ |
| LassoNet | $73.50 \pm 4.64$ | $93.43 \pm 0.41$ | $\mathbf{76.12 \pm 3.80}$ | $89.54 \pm 1.92$ | $74.21 \pm 6.81$ |
| STG | $\underline{79.00 \pm 3.39}$ | $87.22 \pm 3.91$ | $71.08 \pm 2.24$ | $83.84 \pm 3.80$ | $77.89 \pm 3.00$ |
| QS | $73.75 \pm 4.15$ | - | $75.00 \pm 0.00$ | $77.38 \pm 5.19$ | - |
| Fisher | $60.00 \pm 1.58$ | $64.66 \pm 0.18$ | $63.90 \pm 0.00$ | $\mathbf{90.50 \pm 0.00}$ | $\underline{80.53 \pm 2.35}$ |
| CIFE | $50.00 \pm 0.00$ | - | $69.40 \pm 0.00$ | $52.40 \pm 0.00$ | - |
| ICAP | $\mathbf{80.00 \pm 0.00}$ | - | $63.90 \pm 0.00$ | $81.00 \pm 0.00$ | - |
| RFS | $75.00 \pm 0.00$ | - | - | $90.50 \pm 0.00$ | - |
| EntryPrune | $72.50 \pm 9.35$ | $88.47 \pm 0.69$ | $\underline{76.11 \pm 1.52}$ | $\underline{90.48 \pm 0.00}$ | $77.37 \pm 3.99$ |
| EntryPrune flex | $78.00 \pm 6.71$ | $86.02 \pm 0.91$ | $75.00 \pm 3.40$ | $\underline{90.48 \pm 0.00}$ | $\mathbf{82.11 \pm 3.43}$ |
| **Learner: KNN** | | | | | |
| All | $92.50$ | $80.95$ | $69.44$ | $76.19$ | $73.68$ |
| XGBoost | $\underline{77.50 \pm 0.00}$ | $\mathbf{91.98 \pm 0.00}$ | $63.89 \pm 0.00$ | $76.19 \pm 0.00$ | $65.79 \pm 0.00$ |
| SHAP-XGBoost | $75.00 \pm 0.00$ | $82.96 \pm 0.00$ | $\mathbf{77.78 \pm 0.00}$ | $80.95 \pm 0.00$ | $73.68 \pm 0.00$ |
| CancelOut | $56.50 \pm 4.18$ | $78.25 \pm 8.12$ | $43.33 \pm 13.26$ | $81.90 \pm 5.22$ | $66.32 \pm 3.43$ |
| GradEnFS | $57.50 \pm 8.66$ | $88.87 \pm 3.13$ | $52.78 \pm 2.78$ | $80.00 \pm 9.16$ | $66.32 \pm 3.90$ |
| NeuroFS | $74.00 \pm 5.15$ | $88.48 \pm 1.54$ | $64.42 \pm 5.38$ | $85.86 \pm 4.67$ | $\underline{76.82 \pm 2.18}$ |
| LassoNet | $67.50 \pm 7.75$ | $\underline{91.13 \pm 0.46}$ | $\underline{68.90 \pm 4.07}$ | $82.86 \pm 3.80$ | $62.11 \pm 3.00$ |
| STG | $75.00 \pm 5.24$ | $83.16 \pm 2.31$ | $58.90 \pm 7.52$ | $81.00 \pm 0.00$ | $74.74 \pm 3.99$ |
| QS | $75.00 \pm 3.54$ | - | $66.70 \pm 0.00$ | $65.47 \pm 8.37$ | - |
| Fisher | $70.00 \pm 0.00$ | $55.89 \pm 0.00$ | $50.00 \pm 0.00$ | $85.70 \pm 0.00$ | $\mathbf{81.58 \pm 0.00}$ |
| CIFE | $70.00 \pm 0.00$ | - | $44.40 \pm 0.00$ | $57.10 \pm 0.00$ | - |
| ICAP | $65.00 \pm 0.00$ | - | $61.10 \pm 0.00$ | $66.70 \pm 0.00$ | - |
| RFS | $\mathbf{85.00 \pm 0.00}$ | - | - | $\mathbf{90.50 \pm 0.00}$ | - |
| EntryPrune | $73.00 \pm 7.37$ | $87.62 \pm 1.45$ | $58.89 \pm 4.12$ | $87.62 \pm 2.61$ | $71.58 \pm 6.00$ |
| EntryPrune flex | $76.50 \pm 2.24$ | $83.06 \pm 3.51$ | $62.22 \pm 6.09$ | $\underline{89.52 \pm 2.13}$ | $71.05 \pm 3.22$ |
| **Learner: SVM** | | | | | |
| All | $77.50$ | $94.24$ | $72.22$ | $80.95$ | $84.21$ |
| XGBoost | $72.50 \pm 0.00$ | $90.48 \pm 0.00$ | $72.22 \pm 0.00$ | $90.48 \pm 0.00$ | $76.32 \pm 0.00$ |
| SHAP-XGBoost | $\mathbf{77.50 \pm 0.00}$ | $89.47 \pm 0.00$ | $\underline{77.78 \pm 0.00}$ | $\underline{90.48 \pm 0.00}$ | $76.32 \pm 0.00$ |
| CancelOut | $58.50 \pm 2.85$ | $81.70 \pm 5.60$ | $58.89 \pm 6.63$ | $87.62 \pm 4.26$ | $78.42 \pm 1.18$ |
| GradEnFS | $69.00 \pm 7.83$ | $\underline{91.38 \pm 0.91}$ | $64.44 \pm 6.63$ | $87.62 \pm 4.26$ | $77.37 \pm 2.35$ |
| NeuroFS | $76.50 \pm 2.55$ | $88.08 \pm 0.70$ | $\mathbf{80.54 \pm 4.96}$ | $\mathbf{90.50 \pm 0.00}$ | $81.56 \pm 2.65$ |
| LassoNet | $71.00 \pm 2.00$ | $\mathbf{91.98 \pm 1.16}$ | $74.46 \pm 4.78$ | $88.58 \pm 2.35$ | $80.53 \pm 3.99$ |
| STG | $71.00 \pm 2.55$ | $84.41 \pm 3.20$ | $70.00 \pm 4.08$ | $84.78 \pm 3.55$ | $81.58 \pm 1.86$ |
| QS | $74.38 \pm 4.80$ | - | $72.20 \pm 2.80$ | $76.20 \pm 7.53$ | - |
| Fisher | $67.50 \pm 0.00$ | $62.16 \pm 0.00$ | $63.90 \pm 0.00$ | $\mathbf{90.50 \pm 0.00}$ | $78.95 \pm 0.00$ |
| CIFE | $52.50 \pm 0.00$ | - | $58.30 \pm 0.00$ | $47.60 \pm 0.00$ | - |
| ICAP | $70.00 \pm 0.00$ | - | $72.20 \pm 0.00$ | $57.10 \pm 0.00$ | - |
| RFS | $\mathbf{77.50 \pm 0.00}$ | - | - | $\mathbf{90.50 \pm 0.00}$ | - |
| RigL | $\underline{77.00 \pm 3.32}$ | - | $70.54 \pm 4.16$ | $79.06 \pm 7.11$ | - |
| EntryPrune | $72.50 \pm 5.59$ | $86.47 \pm 1.45$ | $73.33 \pm 1.52$ | $90.48 \pm 0.00$ | $\mathbf{83.68 \pm 4.32}$ |
| EntryPrune flex | $76.00 \pm 6.75$ | $84.56 \pm 1.67$ | $74.44 \pm 2.32$ | $\underline{89.52 \pm 2.13}$ | $\underline{82.11 \pm 3.43}$ |

## G  ARCHITECTURE COMPATIBILITY

One contributing factor to the performance gap between EntryPrune and NeuroFS on the most complex dataset studied, CIFAR-100, is their architectural differences. While NeuroFS employs a sparse MLP with three hidden layers of 1000 neurons each, the original EntryPrune configuration used a single hidden layer with only 100 neurons. To assess the impact of architecture on EntryPrune's performance, we investigate its compatibility with a larger model.

Specifically, we evaluate EntryPrune using a deeper architecture with two hidden layers, each containing 1000 neurons. We found that training for 500 epochs and increasing the number of mini-batches before rotation ($n_{\text{mb}} = 1500$) worked well.

The results, shown in Table 7, indicate that EntryPrune is generally compatible with larger architectures, and that increasing model complexity can improve performance on challenging datasets. In this configuration, 'EntryPrune large' outperforms the other methods across most values of $K$.

Table 7: Accuracy for different numbers of selected features $K$ on the CIFAR-100 dataset. The best method for each $K$ is marked in bold. Entries represent the mean $\pm$ standard deviation of SVM downstream accuracy over five runs.

| Method | K=25 | K=50 | K=75 | K=100 | Average |
|---|---|---|---|---|---|
| NeuroFS | **17.40** | 21.10 | 22.10 | 23.20 | 20.95 |
| LassoNet | 9.58 ± 1.05 | 10.55 ± 0.50 | 12.41 ± 2.15 | 13.25 ± 2.22 | 11.45 |
| Fisher | 4.60 | 5.21 | 6.05 | 6.62 | 5.62 |
| EntryPrune | 15.33 ± 0.84 | 18.98 ± 0.90 | 20.40 ± 0.63 | 22.34 ± 0.43 | 19.26 |
| EntryPrune flex | 15.26 ± 1.09 | 19.13 ± 1.34 | 21.52 ± 0.58 | 23.19 ± 0.18 | 19.77 |
| EntryPrune large | 16.82 ± 0.87 | **21.49 ± 0.22** | **23.17 ± 0.31** | **24.17 ± 0.33** | **21.41** |

One architectural limitation of our approach is its incompatibility with networks that use weight sharing in the first layer. This includes convolutional neural networks and transformer-based architectures such as Vision Transformers (Khan et al., 2022). Weight sharing makes it infeasible to estimate the relative impact of individual features using the change-based scoring used by EntryPrune. However, this limitation is shared by other neural network-based approaches such as LassoNet and NeuroFS. Addressing feature selection in the presence of weight sharing remains an important direction for future work.

## H  STOPPING CRITERIA

In this section, we provide an explorative analysis to evaluate the performance of various stopping criteria and hyperparameters. While the main experiment employed a single stopping protocol across all conditions—consistent with the baseline methods for comparison—this exploration highlights feasible parameter ranges and assesses whether fine-tuning stopping rules for specific configurations can lead to improvements.

We analyze three stopping criteria:

- Epochs: The number of training epochs.
- Ident: The number of updates without changes to the identified feature set.
- Validation: A combination of updates without improvements in validation loss and updates without changes to the identified feature set, as used in the main experiment.

**Identifying suitable parameters.**   To evaluate these criteria, we performed an initial analysis on a long dataset (ISOLET) and a wide dataset (ARCENE), using $K = 50$ selected features. The corresponding hyperparameters were varied as follows: Epochs between 1 and 5000, Ident patience between 1 and 400, and Validation patience between 1 and 200.

For each criterion, 30 hyperparameter configurations were tested, selected using Optuna (Akiba et al., 2019) to balance exploration and exploitation. All other settings were consistent with the

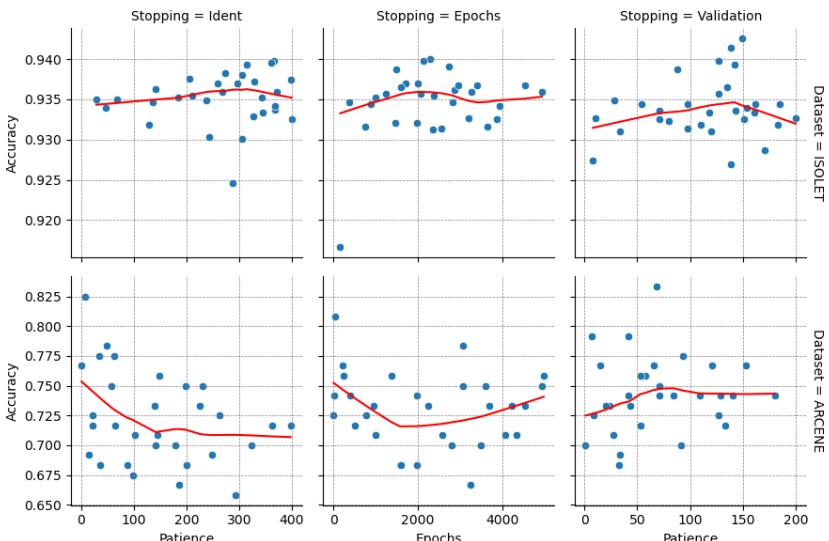

Figure 12: Accuracy by stopping criterion for $K = 50$ selected features. The red line indicates a LOWESS interpolation.

main experiment. In each configuration, the resulting SVM accuracy was averaged over three runs. Figure 12 presents the results by dataset and criterion. The results suggest that all stopping criteria can perform well with appropriately chosen hyperparameters. However, the ident criterion appears highly dataset-dependent and may require fine-tuning using validation data. Similarly, specific epoch values do not generalize well across datasets: for instance, while 2000 epochs performed well for ISOLET, it was suboptimal for ARCENE. The validation criterion, in contrast, demonstrated greater robustness across datasets, with the patience value around 100 yielding consistent performance.

**Assessing variance across configurations.** A second case study investigated the performance variance of stopping rules across different numbers of selected features. The GLA-BRA-180 dataset was chosen for this analysis, as EntryPrune underperformed in the $K = 50$ configuration. Table 8 provides a detailed breakdown of stopping performance on the GLA-BRA dataset for all studied numbers of selected features. Note that, to compare the criteria, the validation criterion with a

Table 8: Accuracy by stopping criterion for different numbers of selected features $K$ for the GLA-BRA-180 dataset. Entries corresponding to the stopping criterion and hyperparameter used in the main experiment are marked in bold. Values represent the mean SVM accuracy across five runs.

| | Validation | | | | | | Epochs | | | | |
|---|---|---|---|---|---|---|---|---|---|---|---|
| K | 50 | **100** | 150 | 200 | 400 | 500 | 250 | 500 | 1000 | 2500 | 5000 |
| 25 | 73.33 | **75.0** | 73.33 | 74.44 | 76.11 | 74.44 | 73.89 | 74.44 | 73.89 | 78.89 | 73.33 |
| 50 | 75.0 | **73.33** | 76.11 | 76.11 | 78.89 | 72.78 | 72.78 | 72.78 | 76.11 | 74.44 | 72.22 |
| 75 | 76.67 | **77.78** | 73.33 | 75.56 | 76.11 | 76.11 | 72.78 | 76.11 | 82.22 | 72.78 | 73.33 |
| 100 | 75.0 | **77.22** | 78.33 | 80.56 | 76.11 | 76.11 | 76.11 | 76.11 | 76.11 | 75.0 | 75.0 |

patience value of 100 corresponded, on average, to approximately 500 epochs. The analysis shows that increasing the validation patience to 400 improves performance for $K = 50$, approaching the state of the art. However, this improvement is not consistent across all $K$ values, as performance stagnates for 75 and 100 selected features. Similarly, no general trend emerges for the Epochs criterion to improve the performance consistently across all $K$. Consequently, while Valdation stopping using a patience of 100 delivers good performance, fine-tuning the stopping parameter using validation data for specific numbers of selected features can further improve results.

# I  LARGE LANGUAGE MODEL USAGE

We utilized Large Language Models (LLMs) to assist with grammar correction, typo fixing, and improving overall text flow. The models were employed solely for polishing language and did not influence the technical content, experimental design, or interpretation of results. This process enhanced clarity and readability while preserving the authors' original contributions.

