# OpenReview forum: "EntryPrune: Neural Network Feature Selection using First Impressions"
_ICLR.cc/2026/Conference — Submitted to ICLR 2026_

### Official Review · Reviewer_osxp · 2025-10-26

**Soundness:** 2
**Presentation:** 2
**Contribution:** 2
**Rating:** 4
**Confidence:** 5

**Summary:**

The authors propose EntryPrune, an embedded feature-selection method built on a dense MLP with a dynamically sparse input layer. The key idea is an entry-based pruning criterion. Specifically, when a pruned input feature is (re)introduced as a candidate into the input layer, the algorithm measures the short-horizon relative change it induces. This is operationalized as the z-scored L1-norm of its first-layer gradient accumulations over the first n mini-batches post-entry. These accumulated scores are stored for all features. Periodically, the algorithm identifies the K features with the largest stored entry scores and prunes the rest. Then, it randomly regrows a batch of new candidate inputs whose incoming weights are reinitialized to tiny values. According to the authors, this strategy balances between established and freshly (re)introduced features and mitigates the bias of magnitude-based pruning toward long-tenured inputs.
Empirically, across 13 datasets (tabular, vision, speech, genomics, and text), EntryPrune often outperforms or matches SotA feature selection baselines (NeuroFS, LassoNet, STG, QS, RFS, etc.). An ablation suggests that entry-based scoring (short-horizon gradient sums) beats magnitude‐based criteria inside the same framework. The proposed method often runs faster wall-clock than a sparse-training baseline (NeuroFS). The code and replication scripts are provided.

**Strengths:**

1. The algorithmic contribution is clear. The entry-based pruning idea, i.e., scoring features by their initial impact upon (re)entry and freezing that score to avoid tenure bias, is straightforward, well motivated, and easy to implement. The random regrowth and tiny reinitialization are thoughtfully chosen to avoid gradient suppression and encourage exploration.
2. The empirical evaluation is nice. The paper evaluates on 13 datasets spanning images, speech, sensor, and genomics, with reports of SVM accuracies and additional downstream models. Various ablations strengthen the analysis.
3. The code is provided; many baselines are taken from public repositories. The paper documents settings and includes extensive appendix tables.

**Weaknesses:**

1. The related work discussion is incomplete. For example, recent approaches, such as "CancelOut: A Layer for Feature Selection in Deep Neural Networks" by Borisov et al., or "Leveraging model inherent variable importance for stable online feature selection" by Haug et al. are not discussed. In fact, I think they should be considered as competitors in the evaluation.
2. Uncertainty quantification plays a critical role when it comes to feature selection. EntryPrune does not consider this aspect. Is there a way to measure/quantify the uncertainty of the selection process in EntryPrune?
3. It is not clear what the contribution of the entry metric from regrowth is. Is it possible to isolate this contribution?
4. The robustness analysis could be much deeper. Would a simple exponential decay or batch-normalized cumulative score improve robustness? In which scenarios does EntryPrune fail? What is the impact of concept shifts or other types of drifts in the data on EntryPrune's performance?

**Questions:**

See comments above.

---

> ### Author Response · Authors · 2025-11-20
> **Response to Reviewer osxp**
>
> We greatly appreciate your constructive feedback and thoughtful comments. We have addressed each of your points in detail below.
>
> ---
>
> ### [W1] Coverage of Related Work & Baselines
>
> Thanks for the suggestion. We have now integrated CancelOut into the related work discussion and included it in the main experiments to provide a stronger comparison.
>
> Regarding “Leveraging model inherent variable importance for stable online feature selection” (Haug et al., 2020), we note that it focuses on online feature selection with streaming data and concept drift. In contrast, EntryPrune operates in the offline setting on static datasets. While conceptually related, this method is not directly comparable in scope or evaluation (see our response to W4).
>
> ---
>
> ### [W2] Uncertainty Quantification in Feature Selection
>
> One way to capture uncertainty in feature selection is by examining how consistently a method selects the same features across independent runs. We assess this aspect of EntryPrune in detail in Appendix C. We welcome any suggestions for additional ways to quantify uncertainty.
>
> ---
>
> ### [W3] Contribution of Entry Metric
>
> We isolate the contribution of entry-based pruning in Section 4.2 (Figure 7), where experiments show it outperforms other classical pruning approaches.
>
> ---
>
> ### [W4] Depth of Robustness Analysis & Failure Modes
>
> We thank the reviewer for these suggestions. Our paper already includes several ablations and analyses that assess different aspects of robustness. We experimented with adding exponential decay to the entry scores during development, but it did not yield consistent improvements, so we focused the ablation studies on the core components (pruning criterion, hyperparameters).
>
> We are not certain what you mean by “batch-normalized cumulative scores.” One possible interpretation could be the use of batch-normalized cumulative gradients instead of the standardized cumulative gradients in Steps 6–10 of Algorithm 1. Another could involve batch-normalized cumulative entry scores, although it is unclear how such normalization would remain compatible with comparisons between newly regrown and long-active features.
>
> Regarding failing scenarios, we perform several investigations on architecture and hyperparameter settings that lead to suboptimal performance. For example, we show that bigger architectures are needed for more complex datasets (Appendix G) and the size of the input layer represents a stability/performance tradeoff (Appendix C).
>
> Regarding robustness to concept shifts and data drift, these aspects are beyond the scope of the present study, which focuses on static, real-world datasets where all data is available from the start. Investigating EntryPrune’s behavior under data stream settings would indeed be a valuable direction for future research. In such contexts, classical options for handling drift, such as exponential decay, would certainly be worth revisiting.
>
> ---
>
> Thank you for your help in improving our work. We hope that these updates address your concerns and are looking forward to your feedback.

---

> > ### Comment · Reviewer_osxp · 2025-11-23
> >
> > I thank the authors for having addressed my main concerns. The extended related work and baselines, as well as the clarified analyses on stability and architectural/hyperparameter sensitivity, go some way toward addressing my questions about robustness and uncertainty. However, a more systematic study of data drift remains open or part of future work. I will increase my score accordingly.

---

### Official Review · Reviewer_Nw9T · 2025-10-29

**Soundness:** 2
**Presentation:** 3
**Contribution:** 3
**Rating:** 2
**Confidence:** 3

**Summary:**

This paper proposes EntryPrune, a feature selection method that combines a dense neural network with a dynamically sparse input layer, where features are iteratively pruned and regrown. The main idea is an entry-based pruning metric that evaluates each new feature based on its early gradient-driven impact. The method is benchmarked on 10+ datasets, and (according to the results in the manuscript) it shows strong performance particularly on datasets with more samples than features.

**Strengths:**

- Novel method
- Good experimental coverage (13 datasets)
- The paper is well written and easy to follow

**Weaknesses:**

- Although the paper motivates feature selection as a path to interpretability, it does not connect its contribution to established explainability methods such as SHAP, LIME, Integrated Gradients, or Grad-CAM. Given that the method relies on gradients and is applied to image data, this omission weakens the interpretability claim

- On wide datasets, the method offers little to no improvement over existing baselines

- The experimental setup relies mainly on SVMs, which are somewhat outdated, incorporating more modern models such as GBDTs or Random Forests would provide a fairer and more relevant comparison

- The evaluation also omits tree-based feature selection baselines (e.g., feature importance from Random Forest or GBDT), which are widely used in practice and should at least be discussed

**Questions:**

- How does EntryPrune compare to attribution-based interpretability methods (e.g., SHAP, LIME, Grad-CAM, ...)? Could the authors clarify whether EntryPrune should be viewed as a competing interpretability approach or a complementary one?
- Please see the "Weaknesses" section

---

> ### Author Response · Authors · 2025-11-20
> **Response to Reviewer Nw9T**
>
> We greatly appreciate your constructive feedback and thoughtful comments. We have addressed each of your points in detail below.
>
> ---
>
> ### [W1] Relation to Explainability Methods
>
> Classical explainability methods provide post-hoc attributions for individual predictions. We view feature selection as a complementary, rather than competing, approach to interpretability: it identifies the minimal set of features required to maintain or improve overall model performance, offering a broader view of interpretability through the lens of performance preservation. This perspective aligns with recent studies showing that selective feature use can enhance predictive performance—for example, “selecting only 4–14% of features can improve heart attack prediction (Akter et al., 2025)” as noted in the Introduction.
>
> ---
>
> ### [W2] Performance on Wide Datasets
>
> On wide datasets, it outperforms on SMK, underperforms on BASEHOCK, and performs similarly to the state of the art on the remaining three datasets. It should be noted that, as shown in Figure 4, the higher standard errors for wide datasets indicate lower separability between methods, with no clear performance gaps compared to the more distinct separation observed for long datasets.
>
> ---
>
> ### [W3] Choice of Downstream Learners
>
> We use SVMs as the downstream learner across all conditions. To evaluate whether results differ with other learners, we also test ExtraTrees (a GBDT method) and KNN for the K = 50 selected features condition. The results are consistent across these learners, as shown in Tables 5 and 6.
>
> ---
>
> ### [W4] Missing Tree-Based FS Baselines
>
> Thanks for the suggestion. We have now integrated XGBoost as a GBDT baseline into the main experiments to provide a stronger comparison.
>
> ---
>
> ### [Q1] Complementarity with Attribution Methods
>
> As noted in our response to W1, we view EntryPrune as complementary to attribution-based methods. While methods like SHAP, LIME, or Grad-CAM provide post-hoc explanations for individual predictions, EntryPrune identifies a minimal set of features required to maintain or improve overall model performance, offering a broader perspective on interpretability.
>
> ---
>
> Thank you for your help in improving our work. We hope that these updates address your concerns and are looking forward to your feedback.

---

> > ### Comment · Reviewer_Nw9T · 2025-11-20
> > **response**
> >
> > Dear Authors, thank you for your detailed answers to my questions, given the new results with tree-based methods I updated the score.
> >
> > However, in my opinion, feature selection and xAI methods serve the same purpose: to find the features or attributes that are the most influential to a chosen target.
> >
> > >While methods like SHAP, LIME, or Grad-CAM provide post-hoc explanations for individual predictions,
> >
> > This is true, however, we can use the above-mentioned methods to create a global explanation (e.g., aggregate local explanations by taking the mean of all local explanations) and then see it a feature selection method.
> >
> > Thus, SHAP/LIME or others could **also** be used to enhance predictive performance. I do suggest including the justification for why xAI methods are not included, or ideally including some of the methods in the experiments.
> >
> >
> > Minor:
> > - Table 3, for k=100 Authors could highlight bold all the best scores, not just the winning method.

---

> > > ### Author Response · Authors · 2025-12-01
> > >
> > > Thank you for the constructive feedback and for engaging with our rebuttal. We agree with your perspective that feature selection and xAI methods share the broader goal of identifying influential features, albeit from different angles. As you note, local attribution methods such as SHAP and LIME can be aggregated into global importance scores, providing a natural way to use them for feature selection.
> > >
> > > Following your suggestion, we have now incorporated SHAP-XGBoost into our experimental comparison, thereby strengthening the evaluation to include xAI-based feature selection.
> > >
> > > Regarding your minor comment on Table 3, thank you for pointing this out. We have corrected the highlighting for all tables to consistently mark all best-performing scores.

---

### Official Review · Reviewer_Jpgm · 2025-10-31

**Soundness:** 2
**Presentation:** 3
**Contribution:** 2
**Rating:** 4
**Confidence:** 4

**Summary:**

This manuscript proposes a supervised feature selection algorithm called EntryPrune. The algorithm is based on a dense neural network with a dynamically sparse input layer. The core mechanism is claimed to be "entry-based pruning", evaluating the importance of a neuron(feature) based on the relative change induced when it first enters the network, combined with a random regrowth strategy. The authors claim that this method outperforms (or matches) sota on 13 datasets, especially on "long" datasets.

**Strengths:**

1. The pruning approach proposed in this paper is an interesting heuristic, which attempt to address the issue of unfair evaluation time between new and old neurons in dynamic sparse training.
2. Compared to NeuroFS and LassoNet, the proposed method may have lower computation time while maintaining comparable performance.

**Weaknesses:**

1. Dynamic sparse training is a widely researched and used approach. The method proposed in this manuscript is more like an incremental improvement on the existing NeuroFS framework. The most creative part is the introduction of a new pruning metric strategy. In addition, this manuscript avoids any theoretical analysis of its effectiveness.

2. The manuscript seems to have ignored GBDT baselines e.g., xgboost and catboost, in the main text, and appendix B also seems to show GBDT's powerful ability to identify interactive features.

3. The manuscript argues that "random regrowth" is superior to "gradient-based regrowth" because the former can discover "interaction features." However, the only evidence for this claim comes from a toy example in Appendix B, lacking studies on more real-world datasets .

4. Fig.11 shows that the feature subset selected by the proposed method has low stability in multiple runs, which means that the reliability of this method may be highly dependent on hyperparameter changes. This is unacceptable in fields such as healthcare and finance.

5. While the method proposed in the manuscript performs well on homogeneous datasets such as images and speech, it performs poorly on many so-called "wide datasets" (where the number of features is greater than the number of samples, such as ARCENE and GLA-BRA-180). The paper's claim of "better overall performance" is inconsistent with the data.

**Questions:**

see weakness.

---

> ### Author Response · Authors · 2025-11-20
> **Response to Reviewer Jpgm**
>
> We greatly appreciate your constructive feedback and thoughtful comments. We have addressed each of your points in detail below.
>
> ---
>
> ### [W1] Novelty vs. NeuroFS & Lack of Theory
>
> Apart from its feature selection purpose, EntryPrune differs from NeuroFS in all key aspects of a dynamic sparse training approach, including pruning, regrowth, and architecture. Beyond the novel pruning metric, EntryPrune uses random regrowth, which is motivated by insights into feature interactions (Appendix B) and only keeps the input layer sparse rather than the full network.
>
> The entry-based pruning mechanism is inherently challenging to analyze theoretically because it alternates between continuous weight optimization (SGD) and discrete, history-dependent mask updates, creating a non-stationary, bi-level optimization problem; these challenges are discussed in detail in Section 3 of the main text.
>
> ---
>
> ### [W2] Missing GBDT Baselines
> Thanks for the suggestion. We have now integrated XGBoost as a GBDT baseline into the main experiments to provide a stronger comparison.
>
> ---
>
> ### [W3] Evidence for Random Regrowth
> We thank the reviewer for this comment. Our claim regarding random regrowth is based on the fundamental limitation of gradient-based regrowth: it cannot reliably distinguish interaction features from unrelated features early on, since both initially produce similar gradients. While establishing ground truth for feature interactions is straightforward in toy settings (as in Appendix B), this distinction is inherently ambiguous in real-world datasets, making it challenging to study reliably.
>
> ---
>
> ### [W4] Stability and Reliability Concerns
> We thank the reviewer for this comment. Appendix C shows that input layer size trades off two aspects of reliability: larger layers improve stability of the selected parameters but can reduce feature quality due to increased noise. In applications such as healthcare and finance, feature quality is also critical—stability alone is not sufficient, since a fixed feature set could trivially maximize stability but miss important signals. Fig. 11 shows that with the hyperparameters used in our main experiments, EntryPrune achieves stability comparable to NeuroFS while maintaining high-quality feature selection.
>
> ---
>
> ### [W5] Performance on Wide Datasets
>
> Overall, EntryPrune outperforms the state of the art on 9 out of 13 datasets, with notable improvements on ISOLET, MNIST, and FASHION-MNIST. On wide datasets, it outperforms on SMK, underperforms on BASEHOCK, and performs similarly to the state of the art on the remaining three datasets. It should be noted that, as shown in Figure 4, the higher standard errors for wide datasets indicate lower separability between methods, with no clear performance gaps compared to the more distinct separation observed for long datasets.
>
> ---
>
> Thank you for your help in improving our work. We hope that these updates address your concerns and are looking forward to your feedback.

---

### Official Review · Reviewer_3ddh · 2025-11-01

**Soundness:** 2
**Presentation:** 2
**Contribution:** 1
**Rating:** 2
**Confidence:** 3

**Summary:**

The paper introduces EntryPrune, a feature selection method for neural networks that uses a dynamically sparse input layer and entry-based pruning. The key idea is to evaluate features based on their initial impact when they first enter the network. The paper reports that EntryPrune outperforms established methods like NeuroFS and LassoNet in terms of accuracy and runtime efficiency.

**Strengths:**

The introduction of entry-based pruning; measuring the initial impact is reasonable and normalization method used in the techniques ensures fair comparison. Results show some advantage in runtime in long datasets and marginal improvement over the existing techniques.

**Weaknesses:**

1.	The core contribution of entry-based pruning is incremental at best. The main parts of the technique, gradient based regrowth and pruning largely borrows from prior works like NeuroFS and RigL. The entry-based pruning technique is simply a minor adaptation rather than a truly novel contribution to the field.
2.	The experimental setup with MLP of 1 hidden layer with 100 neurons and large network containing two layers, is too basic and fails to offer a convincing benchmark for current applicability of the method.
3.	It is unclear how the pruning mechanism work in the case of multi layered networks and whether is it applicable to other than dense layers such as convolutional or residual.
4.	Baselines are too old (the latest is one from 2023). I suggest adding atleast GradEnFS (2024) and more to establish EntryPrune’s place in the current literature landscape.
5.	The results are, at best, marginal and for the challenging dataset (Cifar-100), NeuroFS is performing better than the proposed consistently. Also, the reported accuracy ranges of these experiments (~40% for Cifar-10 and ~20% for Cifar-100) raises concerns about the practical significance of such techniques.
6.	There is a significant lack of practical benefit demonstrated. The paper fails to provide clear real-world scenarios where EntryPrune would offer significant advantages over simpler, well-established feature selection methods other than special cases e.g. for interpretability.

**Questions:**

1.	Can you clearly highlight how EntryPrune differ from the existing works in a more substantial way, beyond the use of early batch scoring ?
2.	Could entry scores after some rotations drift over time?
3.	The ablation in Figure 8 show sensitivity to hyperparameters. Can you provide a principled approach or a rule of thumb to set the hyperparameters?

---

> ### Author Response · Authors · 2025-11-20
> **Response to Reviewer 3ddh (1/2)**
>
> We greatly appreciate your constructive feedback and thoughtful comments. We have addressed each of your points in detail below.
>
> ---
>
> ### [W1] Novelty of Entry-Based Pruning
> Unlike NeuroFS and RigL, EntryPrune does not use gradient-based regrowth. Instead, it relies on a novel insight that random regrowth is more effective for uncovering interaction features, as discussed in Section 2 and Appendix B. In addition, the pruning mechanism fundamentally differs from prior work: EntryPrune leverages information from earlier time-points to guide entry-level pruning decisions—an ability absent in existing methods. Our ablation study in Section 4 shows that this novel pruning strategy outperforms other strategies.
>
> ---
>
> ### [W2] Experimental Architecture Adequacy
>
> We chose a single hidden layer network for the main experiments to match the default architecture of LassoNet, ensuring direct comparability. We updated the paper to now include this: “We employ a single hidden layer neural network with 100 neurons and a ReLU activation function, following the default architecture used in LassoNet.” For more complex architectures, we include a two-layer network with 1000 neurons per layer in Appendix G, demonstrating that EntryPrune can outperform NeuroFS on CIFAR-100 under this setup.
>
> ---
>
> ### [W3] Applicability to Multi-Layer and Non-Dense Networks
> A multi-layered network is tested in Appendix G, where EntryPrune achieves superior performance compared to NeuroFS with larger architectures. EntryPrune operates on the input layer and is compatible with networks where the input layer is dense; downstream layers—whether fully connected, sparse, or residual—do not affect the selection mechanism in principle.
>
> For convolutional layers, parameter sharing prevents unambiguous gradient attribution per input feature, making EntryPrune incompatible—a limitation shared with other feature selection approaches such as LassoNet or NeuroFS (see Appendix G). For residual connections starting at the input layer, defining entry scores is more complex due to multiple gradient paths. While this is an interesting direction, we consider it beyond the scope of the current work and leave it for future research.
>
> ---
>
> ### [W4] Breadth and Recency of Baselines
> We thank the reviewer for the suggestion. We have added three additional baselines—GradEnFS, XGBoost, and CancelOut—to further strengthen our comparisons and better situate EntryPrune within the current literature. Across the datasets where both methods were evaluated, EntryPrune outperforms GradEnFS on 8 out of 9 datasets, achieving an average accuracy of 88.01% compared to 85.85% for GradEnFS, demonstrating its strong performance relative to recent feature selection methods.
>
> ---
>
> ### [W5] Practical Significance of Results
>
> While EntryPrune outperforms the state of the art on 9 out of 13 datasets, we acknowledge that some differences are marginal. Notably, significant improvements are observed on ISOLET, MNIST, and FASHION-MNIST. For CIFAR-100, the smaller architecture used for EntryPrune explains the observed performance gap. As reported in the main text and shown in Appendix G, when using a larger architecture comparable to NeuroFS, EntryPrune outperforms NeuroFS on CIFAR-100.
>
> The reported accuracy ranges reflect our experimental setup, which evaluates feature selection performance across multiple basic downstream learners, demonstrating the broad value of the identified feature sets.
>
> ---
>
> ### [W6] Real-World Utility and Efficiency
>
> Our evaluation follows the standard in the field, focusing on benchmarks of downstream learner performance, as discussed in our response to W5. While EntryPrune shows clear advantages across multiple datasets and downstream learners, we acknowledge that demonstrating real-world applications is valuable and are open to including additional assessments or considering suggested scenarios in future work.
>
> We also note that several “simpler, well-established” methods face practical limitations on modern, larger datasets. During preparation of the Appendix G experiments, we observed that ICAP and CIFE required more than five hours per run on CIFAR-10, which made them impractical to include. Prior work similarly reports that RFS can exceed a 12-hour time limit even on smaller datasets (Atashgahi et al., 2023, https://openreview.net/forum?id=GcO6ugrLKp). In contrast, EntryPrune maintains competitive wall-clock efficiency across both long and wide datasets (Figure 6), making it substantially more practical on contemporary high-dimensional benchmarks.

---

> ### Author Response · Authors · 2025-11-20
> **Response to Reviewer 3ddh (2/2)**
>
> ### [Q1] Key Differences from Prior Methods
>
> In short, EntryPrune introduces entry-based pruning, which binds feature importance to the initial impact a feature has after entering the network, and random regrowth, which allows features to incrementally demonstrate their relevance. This leads to more stable and fair comparisons among features during training (as shown in Figure 3) and differs fundamentally from existing approaches such as magnitude pruning, both in theory and in demonstrated performance.
>
> We highlight the key differences of EntryPrune in Section 3 and the Introduction, and we are happy to clarify or further emphasize these differences in the text if the reviewers have suggestions for the most effective presentation.
>
>
> ---
>
> ### [Q2] Stability of Entry Scores Over Time
>
> The entry scores of features currently in the network increase monotonically. When a candidate feature’s score exceeds the minimum among the established features, it replaces the lowest-scoring feature, causing that entry score to update to a higher value. As shown in Figure 2, this process stabilizes over time, and the set of top features becomes increasingly stable, preventing uncontrolled drift.
>
> ---
>
> ### [Q3] Hyperparameter Sensitivity & Tuning Guidance
>
> Yes, as we discuss in Section 4.2, some configurations work better for long versus wide datasets. In our updated paper, we formulated this into an explicit rule of thumb: “In summary, long datasets perform best with low c_ratio and high n_mb, while wide datasets perform well with moderate c_ratio and low n_mb; the configurations used in our experiments provide practical starting points for tuning.” Thank you for this suggestion.
>
> ---
>
> Thank you for your help in improving our work. We hope that these updates address your concerns and are looking forward to your feedback.

---

### Author Response · Authors · 2025-11-20
**Global Response**

We thank the reviewers for their constructive comments.

We are encouraged by the recognition of EntryPrune as a “novel method” (Nw9T) that is “straightforward, well motivated, and easy to implement” (osxp). Reviewers highlighted the “good experimental coverage” (Nw9T) across 13 diverse datasets, noting that the method ensures “fair comparison” (3ddh) and offers an “advantage in runtime” (3ddh) while maintaining “comparable performance” (Jpgm).

Taking into account the reviewers’ feedback, we have made a key improvement to the paper (highlighted in cyan): we added three additional baseline methods suggested by the reviewers (XGBoost, CancelOut, GradEnFS), bringing the total to 12 baselines. These new results further strengthen our experimental analysis, and the overall conclusions regarding EntryPrune’s performance remain unchanged.

We hope these updates address the reviewers' concerns and remain open to further feedback.

---

### Author Response · Authors · 2025-12-01
**Summary of the Discussion Period**

We are very grateful for the additional feedback and acknowledgement provided during the discussion period by Reviewers Nw9T and osxp, and we are glad that the clarifications and revisions made brought the paper into the range of acceptance in their view (2->6, 4->6).

Complementing the edits alongside our first responses, the discussion phase resulted in the addition of another baseline method from the field of xAI, SHAP-XGBoost, which strengthens the comparison further.

We were unaware of the recent de-anonymization leak and can assure that we did not use any leaked information in any way. We regret that this situation may have limited the opportunity for Reviewers 3ddh and Jpgm to respond to our rebuttal and further refine our work based on our interaction.

---

### Meta-Review · Area_Chair_ApaD · 2026-01-08

**Summary:**

This manuscript introduces EntryPrune, an embedded feature-selection method based on a dynamically sparse input layer and an entry-based pruning criterion that scores features by their short-horizon impact upon (re)entry. Reviewers broadly agreed that the algorithm is clearly described, easy to implement, and empirically well studied across a wide range of datasets. The additional baselines and analyses added during rebuttal—including tree-based methods and xAI-inspired feature selection—significantly strengthened the experimental positioning, and several reviewers revised their assessments upward, noting competitive performance and favorable runtime in many settings.

Nevertheless, substantial concerns remain regarding the conceptual depth and generality of the contribution. Multiple reviewers viewed the core idea as an incremental refinement of existing dynamic sparse training and feature-selection frameworks rather than a fundamentally new approach, with limited theoretical grounding. Performance gains were often marginal and inconsistent, particularly on wide or more challenging datasets, and questions persist about robustness, stability, and uncertainty in the selected feature sets. While the work is carefully executed and has practical merit, the current evidence does not yet support the level of novelty and reliability expected for acceptance. With a clearer articulation of what is fundamentally new, deeper robustness analysis, and stronger justification of practical impact, this work could form a solid basis for a future submission.

**Reviewer Concerns:**

Reviewer osxp: clear algorithm and good evaluation; requested stronger related-work/baselines, uncertainty and robustness discussion; after additions and clarifications, increased score, while noting drift analysis remains future work.

Reviewer 3ddh: novelty seen as incremental; architectures too basic; unclear extensibility beyond dense inputs; asked for newer baselines and clearer practical relevance; authors added baselines and clarified scope/architectures.

Reviewer Jpgm: incremental over NeuroFS; no theory; questioned random regrowth justification beyond toy setting; raised stability issues and mixed results on wide datasets; authors added GBDT baseline and clarified stability/claims.

Reviewer Nw9T: strong interest in the method and experiments; asked to connect to xAI and add tree-based baselines; after added tree/xAI baselines and clarifications, increased score.

**Reviewer Scores:**

Reviewer osxp (4): likely around 6  (after expanded baselines and clarifications).

Reviewer 3ddh (2): likely around 2-4 (concerns about novelty/applicability may partially remain despite added baselines and larger-architecture appendix results).

Reviewer Jpgm (4): likely stay 4 (borderline; stability and wide-dataset concerns persist).

Reviewer Nw9T (2): likely around 6 (after additions of tree/xAI comparisons).

---

### Decision · Program_Chairs · 2026-01-26

Reject